# Exploring the virulence gene interactome with CRISPR/dCas9 in the human malaria parasite

Jessica M Bryant[1,2,3,*] (ID), Sebastian Baumgarten[1,2,3] (ID), Florent Dingli[4] (ID), Damarys Loew[4] (ID), Ameya Sinha[5,6] (ID), Aurélie Claës[1,2,3], Peter R Preiser[5,6] (ID), Peter C Dedon[6,7] (ID) & Artur Scherf[1,2,3] (ID)

## Abstract

Mutually exclusive expression of the *var* multigene family is key to immune evasion and pathogenesis in *Plasmodium falciparum*, but few factors have been shown to play a direct role. We adapted a CRISPR-based proteomics approach to identify novel factors associated with *var* genes in their natural chromatin context. Catalytically inactive Cas9 ("dCas9") was targeted to *var* gene regulatory elements, immunoprecipitated, and analyzed with mass spectrometry. Known and novel factors were enriched including structural proteins, DNA helicases, and chromatin remodelers. Functional characterization of *Pf*ISWI, an evolutionarily divergent putative chromatin remodeler enriched at the *var* gene promoter, revealed a role in transcriptional activation. Proteomics of *Pf*ISWI identified several proteins enriched at the *var* gene promoter such as acetyl-CoA synthetase, a putative MORC protein, and an ApiAP2 transcription factor. These findings validate the CRISPR/dCas9 proteomics method and define a new *var* gene-associated chromatin complex. This study establishes a tool for targeted chromatin purification of unaltered genomic loci and identifies novel chromatin-associated factors potentially involved in transcriptional control and/or chromatin organization of virulence genes in the human malaria parasite.

**Keywords** chromatin; CRISPR; epigenetics; *Plasmodium falciparum*; *var* genes
**Subject Categories** Chromatin, Transcription & Genomics; Microbiology, Virology & Host Pathogen Interaction
**Mol Syst Biol. (2020) 16: e9569**

## Introduction

In the malaria parasite *Plasmodium falciparum*, antigenic variation is key to evasion of the immune system and persistence of infection in the human host. The *P. falciparum* variant surface antigen erythrocyte membrane protein 1 (PfEMP1) is a key component in this process and is encoded by the ~ 60-member *var* gene family in the haploid genome (reviewed in (Scherf *et al*, 2008)). A system of mutually exclusive expression is used to activate only a single *var* gene at a time, but occasional switching occurs to facilitate host immune system evasion. Unlike that in other parasites, antigenic variation in *P. falciparum* is under epigenetic control (reviewed in (Cortés & Deitsch, 2017)). The single active *var* gene is associated with euchromatin while all other *var* genes are kept transcriptionally silent via heterochromatin. Functional studies of orthologous histone writers, readers, and erasers have implicated several chromatin-associated proteins in mutually exclusive *var* gene transcription including heterochromatin protein 1 (HP1), the histone deacetylases HDA2 and silent information regulator 2 (SIR2a and b), and the histone methyltransferases SET2 and SET10 (Freitas-Junior *et al*, 2005; Flueck *et al*, 2009; Pérez-Toledo *et al*, 2009; Tonkin *et al*, 2009; Volz *et al*, 2012; Jiang *et al*, 2013; Coleman *et al*, 2014; Ukaegbu *et al*, 2014). Across the intraerythrocytic developmental cycle (IDC) and other stages, HP1 facilitates transcriptional silencing of all but one *var* gene via binding to trimethylation of histone H3 at lysine 9 (H3K9me3) (Flueck *et al*, 2009; Lopez-Rubio *et al*, 2009; Pérez-Toledo *et al*, 2009; Fraschka *et al*, 2018; Zanghì *et al*, 2018). The histone modifications H3K9ac and H3K4me2/3 and the histone variants H2A.Z and H2B.Z were shown to be enriched at the active *var* gene promoter (Lopez-Rubio *et al*, 2007; Petter *et al*, 2013), but the molecular machinery involved in *var* gene activation, such as histone-modifying enzymes or nucleosome remodelers, has yet to be elucidated.

These epigenetic regulatory proteins may be recruited via DNA regulatory elements. All *var* genes have the same basic genetic structure: a 5′ upstream promoter followed by exon I (encodes the polymorphic extracellular domain of the PfEMP1), a relatively conserved intron, and exon II (encodes the conserved intracellular domain; Fig 1A). Both the promoter and the intron have been implicated in mutually exclusive transcription control (reviewed in (Guizetti & Scherf, 2013)). Early studies suggested that the *var* gene

1 Biology of Host-Parasite Interactions Unit, Institut Pasteur, Paris, France
2 INSERM U1201, Paris, France
3 CNRS ERL9195, Paris, France
4 Institut Curie, PSL Research University, Centre de Recherche, Mass Spectrometry and Proteomics Facility, Paris, France
5 School of Biological Sciences, Nanyang Technological University, Singapore, Singapore
6 Antimicrobial Resistance Interdisciplinary Research Group, Singapore-MIT Alliance for Research and Technology, Singapore, Singapore
7 Department of Biological Engineering, Massachusetts Institute of Technology, Cambridge, MA, USA
*Corresponding author. Tel: +33 1 45 68 86 22; E-mail: jessica.bryant@pasteur.fr

promoter could drive transcription unless paired with a downstream *var* gene intron, which had a repressive effect that was purportedly due to its own bidirectional promoter activity (Calderwood *et al*, 2003; Gannoun-Zaki *et al*, 2005; Dzikowski *et al*, 2006, 2007; Frank *et al*, 2006; Epp *et al*, 2009). Conversely, long non-coding RNAs (lncRNA) originating from the *var* gene intron (Fig 1A) have been implicated in the transcriptional activation of *var* genes (Amit-Avraham *et al*, 2015). While deletion of the endogenous *var* gene intron did not block transcriptional silencing or activation of the targeted *var* gene, it did lead to higher rates of *var* gene switching (Bryant *et al*, 2017). Thus, investigating the protein cohort that binds to either the *var* gene promoter or intron could provide new mechanistic insight into *var* gene regulation.

Coordination of the *var* genes, located in subtelomeric (*upsA* and *upsB* type) or central chromosomal (*upsC* type) arrays across thirteen chromosomes (Fig 1B), is likely achieved via spatial positioning within the nucleus. Regardless of genome location, *var* genes form heterochromatic clusters at the nuclear periphery, with the single active *var* gene spatially separated from the rest (Duraisingh *et al*, 2005; Ralph *et al*, 2005; Lopez-Rubio *et al*, 2009; Lemieux *et al*, 2013; Ay *et al*, 2014). This clustering may also facilitate the recombination observed among the polymorphic members of the *var* gene family during the asexual replicative cycle, which generates antigenic diversity via the formation of chimeric genes (Bopp *et al*, 2013; Claessens *et al*, 2014). While the RecQ DNA helicase *Pf*WRN was shown to be important for promoting genome stability and preventing aberrant recombination between *var* genes, the molecular mechanism behind normal mitotic recombination of *var* gene members has not been elucidated (Claessens *et al*, 2018). The telomere repeat-binding zinc finger protein (TRZ) and some members of the DNA/RNA-binding Alba family of proteins or the Apetala2 (ApiAP2) family of transcription factors have been shown to bind to telomeres, subtelomeric regions, or potential *var* gene regulatory elements; however, a direct role for these proteins in transcriptional control, organization, or recombination of *var* genes has not been demonstrated (Flueck *et al*, 2010; Zhang *et al*, 2011; Chêne *et al*, 2012; Goyal *et al*, 2012; Bertschi *et al*, 2017; Martins *et al*, 2017; Sierra-Miranda *et al*, 2017). In addition, actin has been linked to the nuclear positioning and thus transcriptional regulation of *var* genes via binding to the conserved *var* gene intron (Zhang *et al*, 2011). Importantly, proteins such as CCCTC binding factor (CTCF) or lamins, involved in higher-order chromosome organization in metazoans, are absent in *P. falciparum* (Batsios *et al*, 2012; Heger *et al*, 2012).

Most studies of *var* gene biology have relied on *in silico* identification of factors found to be important for epigenetic regulation in other eukaryotic systems; however, *P. falciparum* has already shown important differences to other model organisms with regard to epigenetics (reviewed in (Cortés & Deitsch, 2017)). In addition, most studies investigating proteins that bind to specific DNA motifs have used oligo-based *in vitro* approaches such as gel shift or immunoprecipitation. Thus, a new, unbiased approach is needed to identify novel *var* gene-interacting factors that contribute to transcriptional regulation and organization of *var* genes.

To this end, we adapted a recently developed CRISPR technology for isolating specific genomic loci with a tagged, nuclease-deficient Cas9 (dCas9) (Fujita & Fujii, 2013; Waldrip *et al*, 2014; Liu *et al*,

2017; Myers *et al*, 2018). Importantly, this technique identifies interactions taking place in a natural spatio-temporal chromatin context and does not require genetic modification of the targeted locus. In this study, we show specific dCas9 enrichment at targeted *var* gene DNA regulatory elements with chromatin immunoprecipitation followed by sequencing (ChIP-seq). dCas9 immunoprecipitation followed by liquid chromatography-mass spectrometry (IP LC-MS/MS) confirmed previously identified *var* gene interactors and revealed enrichment of several novel chromatin-associated factors possibly involved in transcriptional control and/or chromatin organization of *var* genes. Using inducible knockdown, ChIP-seq, and proteomics, we functionally characterize the promoter-associated putative nucleosome remodeler, *Pf*ISWI, and demonstrate its role in *var* gene transcriptional activation.

## Results

### Targeting dCas9 to *var* genes

To identify factors associated with putative *var* gene DNA regulatory elements in an unbiased manner, we adapted the CRISPR/dCas9-based engineered DNA-binding molecule-mediated chromatin immunoprecipitation (enChIP) method developed in Fujita and Fujii (2013) to *P. falciparum*. As both the *var* promoter and intron likely contain regulatory protein-binding elements, a tagged dCas9 was directed by an optimized single guide RNA (sgRNA) to either feature (Fig 1A). As a consequence of (i) the need for a GC-rich sequence and (ii) the homology of the promoter and intronic sequences across the *var* gene family, we designed an sgRNA for either genetic feature that targets multiple *var* loci. For the promoter sgRNA, we identified a 20-base-pair (bp) sequence flanking a protospacer adjacent motif (PAM) that was ~ 40 bp upstream of the translation start site of 19 *var* genes (Table 1, Table EV1, and Fig EV1A). For the intron sgRNA, we identified a 20-bp sequence flanking a PAM that was in region III (Calderwood *et al*, 2003) of the intron of 13 *var* genes (Table 1, Table EV2, and Fig EV1C). For the promoter and intron sgRNAs, the target sequences were exclusive to *upsA/B*, *upsB*, *upsB/C*, and *upsC* type *var* genes (Fig 1B and Tables EV1 and EV2). There was only a single off-target sequence predicted for the intron sgRNA in the upstream region of PF3D7_0400200, which is a *var* pseudogene with a *var* exon II-like sequence.

To create *P. falciparum* strains expressing *var* gene-targeted dCas9, we generated two types of plasmids—pUF1_dCas9-3HA and pL7_*var*_IP—based on the two-plasmid system originally designed for CRISPR/Cas9 in *P. falciparum* (Ghorbal *et al*, 2014). pUF1_dCas9-3HA leads to expression of a hemagglutinin-tagged (HA) catalytically inactive Cas9 containing the RuvC and HNH mutations originally described in Qi *et al* (2013). The pL7 plasmid encodes a sequence-optimized sgRNA described in Dang *et al* (2015). A control plasmid (pL7_Control_IP) was used to determine background dCas9 binding to the genome, as it generates a non-specific sgRNA that does not have a predicted target site in the *P. falciparum* 3D7 genome (Table 1). A bulk culture of 3D7 parasites was transfected with pUF_dCas9-3HA and either pL7_Control_IP, pL7_*var*Promoter_IP, or pL7_*var*Intron_IP. Western blot analysis of the cytoplasmic, nuclear (non-chromatin), and chromatin fractions showed that targeted dCas9 was most enriched in

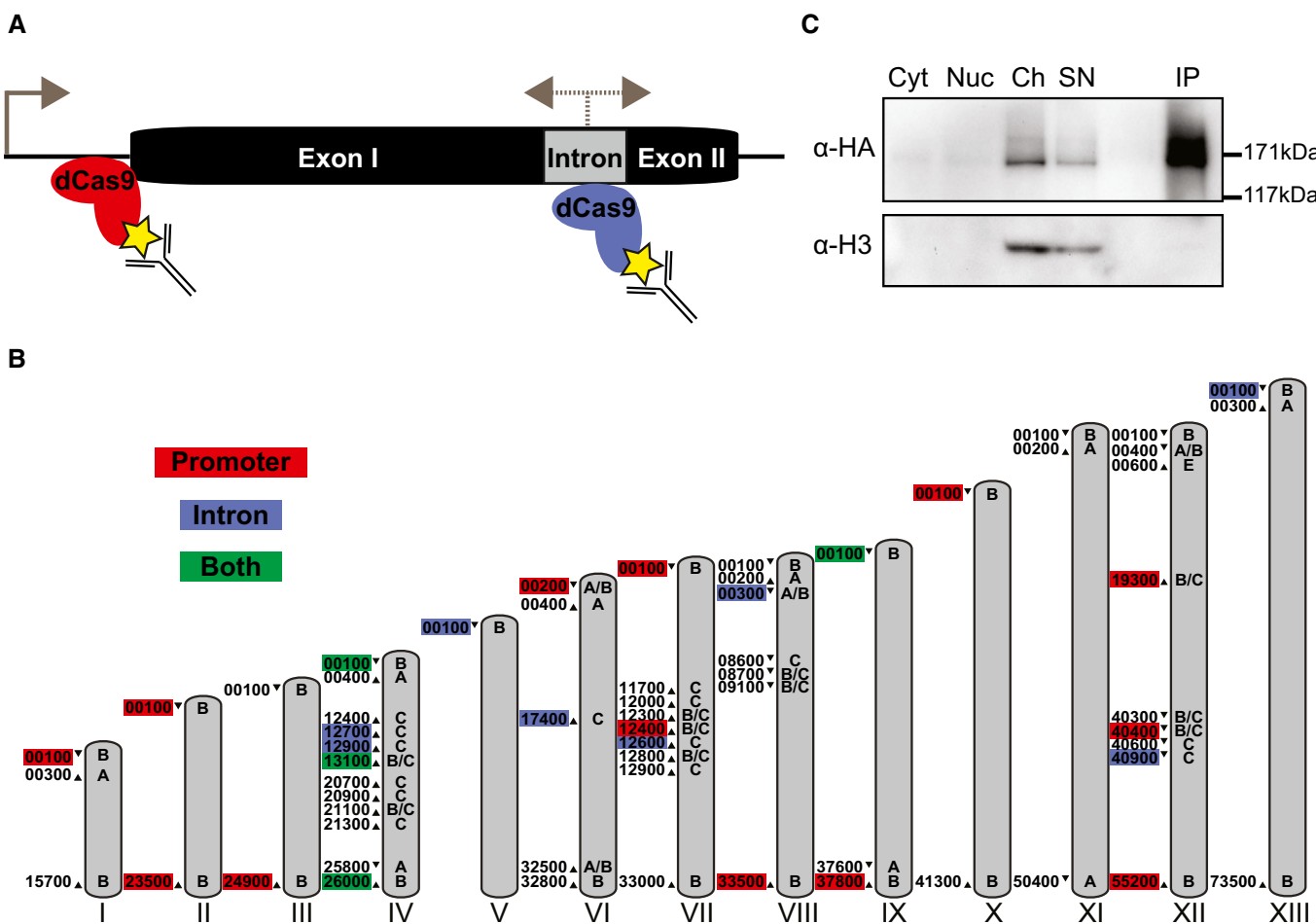

**Figure 1. Immunoprecipitation of *var* gene-targeted dCas9.**

A Schematic of a representative *var* gene with two exons flanking an intron. Transcription originates from the promoter (sense) and intron (sense and antisense). Specific sgRNAs direct dCas9 to either the promoter region (red) or intron (blue). Antibodies were used to isolate the dCas9 and bound genomic regions via a 3xHA tag (yellow star).

B Schematic of *var* genes throughout the *Plasmodium falciparum* genome targeted with intron- (blue), promoter- (red), or both intron and promoter-targeted (green) dCas9. Chromosomes are represented with gray bars, and chromosome numbers are indicated in roman numerals. *var* gene *ups* type is indicated on the chromosome, and *var* gene ID (excluding the preceding chromosome number) is listed to the left of its position on the chromosome. Direction of *var* gene transcription is indicated with an arrowhead.

C Western blot analysis of a dCas9 immunoprecipitation experiment in the promoter-targeted strain at ring stage. Levels of dCas9 and histone H3 in the cytoplasmic (Cyt), nuclear (excluding chromatin, Nuc), and chromatin (Ch) fractions are revealed with anti-HA and anti-H3 antibodies, respectively. dCas9 is enriched in the immunoprecipitated fraction (IP) compared to the unbound supernatant (SN) and input (i.e., chromatin fraction) of the IP. Molecular weights are shown to the right.

Source data are available online for this figure.

the chromatin fraction, and immunoprecipitation of dCas9 from the chromatin fraction with an anti-HA antibody resulted in a robust enrichment of the tagged dCas9 (Fig 1C). The non-targeted dCas9 was present in the nuclear and chromatin fractions (Fig EV1D).

### Genome-wide binding of intron- and promoter-targeted dCas9 is specific and robust

To investigate the specificity of *var* gene-targeted dCas9 binding, we performed ChIP-seq in synchronized parasites cross-linked at 14 h post-invasion (hpi) of the host red blood cells, the ring stage during which *var* gene transcription is highest. Analysis of the ChIP-seq

data revealed that *var* promoter and intron sgRNAs led to specific and significant dCas9 enrichment at most intended target sequences (Fig 2A and Tables EV1 and EV2) while the non-targeted dCas9 control showed no specific enrichment (Fig EV2A). For the promoter sgRNA, a significant peak of dCas9 was found at 17 out of 19 predicted target sites, with only one major unpredicted off-target binding event at PF3D7_1209900 (Figs 2A and EV1B, and Table EV1). For the intron sgRNA, a significant peak of dCas9 was found at 13 out of 16 predicted target sites and the predicted off-target sequence PF3D7_0400200 (Fig 2A and Table EV2). Two additional peaks of dCas9 were detected with the *var* intron-targeted sgRNA at unpredicted *upsB var* gene introns (PF3D7_0500100 and

Table 1. sgRNA characteristics for dCas9 targeting. Shown are the sequences (5′–3′), targeted DNA strand, GC content, and number of on-target (intended) and off-target sequences predicted for each sgRNA.

| sgRNA | Sequence | Target strand | GC content | Predicted On-target | Predicted Off-target |
|---|---|---|---|---|---|
| Promoter | GGTTTATATCGTGGCATACA | Coding | 40% | 19 | 0 |
| Intron | GTAAATGTGTATGTAGTTAT | Coding | 25% | 16 | 1 |
| Control | GCTCGCGATGCTGCCCGACA | — | 70% | 0 | 0 |

PF3D7_1300100) with similar sequences (one or two mismatches; Figs 2A and EV1C, and Table EV2).

Comparison of the ChIP-seq data of the *var* gene-targeted dCas9 and non-targeted dCas9 control highlights the specificity of the method from the most statistically significant dCas9 peak (enrichment ratio = 123.9, $q = 2.22 \times 10^{-289}$ at the promoter of PF3D7_0413100) to the least significant dCas9 peak (enrichment ratio = 42.5, $q = 8.37 \times 10^{-4}$ at the intron of PF3D7_1240900; Fig 2B and Tables EV1 and EV2). dCas9 enrichment was approximately two-fold higher at promoter sgRNA-targeted sites (average ChIP/Input = 20.18) than intron sgRNA-targeted sites (average ChIP/Input = 8.9; Tables EV1 and EV2). However, because the *var* gene intronic sequence is AT-rich relative to the *var* gene promoter sequence (Table 1 and Fig EV1A and C), it could be under-represented in the sequencing data. Regardless, the dCas9 ChIP-seq protocol we present here could be used to determine genome-wide specificity and efficacy of sgRNAs for use in traditional CRISPR/Cas9 genome editing.

To determine the *var* gene transcriptional profile in the strains expressing dCas9, we performed sequencing of mRNA (RNA-seq) from clones and non-clonal bulk parasite cultures at 14 hpi. In clones of the intron- and promoter-targeted dCas9 strains, a single *var* gene was predominantly transcribed, suggesting that mutually exclusive expression of the *var* gene family was unaffected by dCas9 binding (Fig EV2B). RNA-seq of non-clonal bulk parasite cultures of the promoter-, intron-, and non-targeted dCas9 strains showed synchronicity at 14 hpi (Fig EV2C) and that multiple *var* genes, both targeted and untargeted, were transcribed in the population (Fig EV2D). mRNA from the *var* genes targeted at their promoters with dCas9 was, in general, present at higher levels in the promoter-targeted dCas9 strain than in the non-targeted dCas9 strain, suggesting that parasites in which dCas9 was able to more readily bind to the euchromatic promoter were selected for during transfection (Fig EV2D, top). To determine if mRNA was transcribed from *var* genes bound by dCas9 at their promoter, we performed RNA immunoprecipitation (RIP) with dCas9 in the promoter-targeted dCas9 strain in ring stage parasites. Indeed, RNA transcribed from *var* genes bound by dCas9 at their promoters was highly enriched in the promoter-targeted dCas9 RIP (Figs 2C and EV2E, Table EV3).

### Proteomic analysis of dCas9 identifies known and new *var* gene-associated factors

#### Validation of the enChIP method

Having demonstrated specificity of dCas9 binding with ChIP-seq, we set out to identify proteins that associate with either the *var* gene promoter or intron. We performed dCas9-3HA IP LC-MS/MS with the *var* gene promoter- and intron-targeted dCas9 and non-targeted dCas9 strains at 14 hpi. A label-free quantification approach allowed

for comparison of protein enrichment between each sample using low amounts of input material. This led to the quantification of 1,413 total *P. falciparum* proteins (false discovery rate [FDR] of 1%, number of peptides used ≥ 3; Source Data for Tables EV5–8], with dCas9 being the most abundant protein in the LC-MS/MS analysis for each immunoprecipitation. While dCas9 levels were similar in the *var* promoter- and intron-targeted immunoprecipitations (promoter/intron ratio = 1.04, $P = 0.15$; Figs 3A and B, EV3A, Table EV4), dCas9 levels were significantly higher in the non-targeted control when compared to the intron- (ratio = 3.16, $P = 1.88 \times 10^{-62}$) or promoter-targeted dCas9 samples (ratio = 3.07, $P = 5.64 \times 10^{-86}$; Figs 3B and EV3A, Table EV4). Because we found that the number of dCas9 peptides correlated with total proteins identified (Fig EV3A, Table EV4), the non-targeted dCas9 served as an extremely stringent control of background binding.

Thus, we first compared the promoter- and intron-targeted dCas9 data sets for a more unbiased analysis (see Materials and Methods for details of analysis). For most *var* genes, the promoter region is several thousand base pairs upstream of the intron. Thus, our method provides enough resolution to separate factors specifically bound to the promoter or intron, as the average DNA fragment length for the immunoprecipitation was 2 kilobases (Fig EV3B). There were 151 significantly enriched proteins (ratio ≥ 1.5, adjusted *P*-value ≤ 0.05) in the *var* promoter-targeted dCas9 immunoprecipitation (Table EV5) and 115 in the *var* intron-targeted dCas9 immunoprecipitation (Table EV6). Providing strong validation that our method was successful, the AP2 domain-containing SPE2-interacting protein (SIP2, PF3D7_0604100) was the most highly enriched *var* promoter-bound protein in the comparison with the intron-targeted dCas9 (ratio = 11.94, $P = 0.016$, Fig 3A, Table EV5) and the second most highly enriched in the comparison with the non-targeted dCas9 control (ratio = 3.93, $P = 0.02$, Fig 3B, Table EV7). SIP2 was shown to bind to SPE2 DNA motifs upstream of subtelomeric *var* genes, although the role of this protein has not been elucidated (Flueck *et al*, 2010). Indeed, the promoter-targeted dCas9 binds to subtelomeric *var* genes with upstream SPE2/SIP2 sites (Fig EV3C, Table EV1).

When compared to the promoter-targeted dCas9, the intron-targeted dCas9 was enriched in the Alba DNA/RNA-binding proteins 1–3 (ratio = 1.57, $P = 6.6 \times 10^{-4}$ for PF3D7_0814200; ratio = 1.52, $P = 2.4 \times 10^{-5}$ for PF3D7_1346300; ratio = 1.62, $P = 4.22 \times 10^{-6}$ for PF3D7_1006200; Fig 3A, Table EV6). The members of the Alba family have been shown to localize to the nuclear periphery and bind to subtelomeric regions as well as a *var* intronic sequence (Zhang *et al*, 2011; Chêne *et al*, 2012; Goyal *et al*, 2012). In addition to chromatin-associated proteins, the intron-targeted dCas9 sample was enriched in several cytoskeletal proteins such as a putative actin-related protein ARP4a (PF3D7_1422800, ratio = ∞),

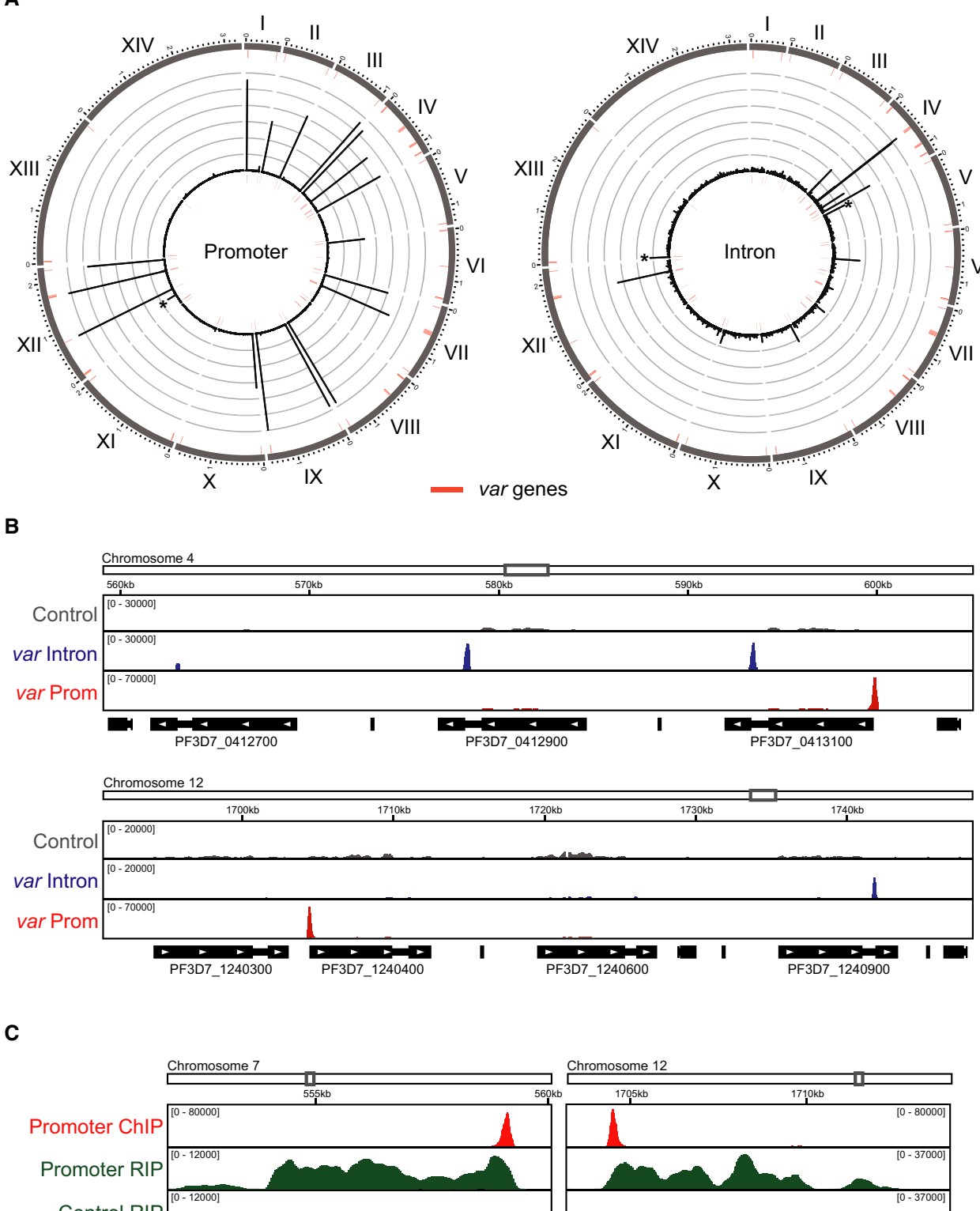

**Figure 2.**

**Figure 2. ChIP sequencing shows specific enrichment of dCas9 at targeted *var* gene introns and promoters.**

A  Circos plots of ChIP-seq data showing genome-wide enrichment of dCas9 in ring stage parasites. The 14 chromosomes are represented circularly by the outer gray bars, with chromosome number indicated in roman numerals and chromosome distances indicated in Arabic numerals (Mbp). Enrichment for intron- or promoter-targeted dCas9 (normalized to non-targeted dCas9) is shown as average reads per million (RPM) over bins of 1,000 nt. The maximum $y$-axis value is 3,000 RPM for the promoter-targeted dCas9 (rings represent increments of 500) and 400 RPM for the intron-targeted dCas9 (rings represent increments of 66.7). *var* genes are represented by red bars. An asterisk indicates an unintended binding event. One replicate was performed for each strain. Peak quantification for promoter- and intron-targeted dCas9 ChIP-seq can be found in Tables EV1 and EV2, respectively.

B  ChIP-seq data show enrichment of dCas9 in strains at ring stage expressing non-targeted "Control" (gray), *var* intron-targeted (blue), or *var* promoter-targeted (red) sgRNAs. Genome location is indicated at the top of each panel. The $x$-axis is DNA sequence, with genes represented by black boxes indented to delineate introns and labeled with white arrowheads to indicate transcription direction. The $y$-axis is input-subtracted ChIP enrichment. $q$-values for promoter-targeted dCas9 peaks shown are $2.22 \times 10^{-289}$ for PF3D7_0413100 and $7.07 \times 10^{-257}$ for PF3D7_1240400. $q$-values for intron-targeted dCas9 peaks shown are $2.03 \times 10^{-76}$ for PF3D7_412700, $6.81 \times 10^{-46}$ for PF3D7_0412900, $1.53 \times 10^{-62}$ for PF3D7_0413100, and $8.37 \times 10^{-4}$ for PF3D7_1240900. One replicate was performed for each strain. Peak quantification for promoter- and intron-targeted dCas9 ChIP-seq can be found in Tables EV1 and EV2, respectively.

C  ChIP-seq (red) and RIP-seq (green) data show enrichment of DNA and RNA, respectively, from *var* genes in the promoter-targeted and non-targeted "Control" dCas9 immunoprecipitation from non-clonal bulk population of ring stage parasites. Genome location is indicated at the top of each panel. The $x$-axis is DNA sequence, with genes represented by black boxes indented to delineate introns and labeled with white arrowheads to indicate transcription direction. The $y$-axis is input-subtracted dCas9 ChIP enrichment in the *var* gene promoter-targeted dCas9 strain (red), dCas9 RIP enrichment normalized to IgG control in the *var* gene promoter-targeted dCas9 strain or non-targeted "Control" dCas9 strain (green), or *var* gene transcript levels in the promoter-targeted dCas9 strain (RPKM, gray). One replicate was performed for ChIP-seq and RNA-seq, and one replicate was performed for HA and IgG control RIP-seq. ChIP-seq peak quantification can be found in Table EV1, and RIP-seq quantification can be found in Table EV3.

myosin A (PF3D7_1342600, ratio = 1.58, $P = 7.65 \times 10^{-5}$), coronin (PF3D7_1251200, ratio = 1.78, $P = 4.76 \times 10^{-5}$), and profilin (PF3D7_0932200, ratio = 1.52, $P = 2.57 \times 10^{-2}$; Fig 3A, Table EV6). The same trend of cytoskeletal protein enrichment can be seen in the intron- versus non-targeted dCas9 comparison (Table EV8). Gene Ontology (GO) analysis showed that proteins that were significantly enriched in the intron-targeted dCas9 immunoprecipitation compared to both the promoter- and non-targeted dCas9 immunoprecipitations are significantly represented by the biological function category of "cell motility" ($P = 0.0018$, Table EV9). Indeed, actin has been implicated in localizing *var* genes to the nuclear periphery by binding to the *var* gene intron (Zhang *et al*, 2011). Thus, our methodology was validated by the identification of specific proteins previously shown to bind to *var* gene upstream regions or introns such as SIP2 and the Alba family.

### The *var* promoter is enriched in DNA replication and repair machinery

Gene Ontology analysis also showed that proteins that were significantly enriched in the promoter-targeted dCas9 immunoprecipitation compared to both the intron- and non-targeted dCas9 immunoprecipitations were significantly represented by the biological function categories of "DNA replication initiation" ($P = 1.77 \times 10^{-5}$) and "DNA replication" ($P = 1.3 \times 10^{-4}$; Table EV10). When compared to the intron- or non-targeted dCas9 samples, the promoter-targeted dCas9 sample was significantly enriched in proteins involved in DNA replication and repair such as DNA helicase 60 (PF3D7_1227100), minichromosome maintenance protein complex subunits MCM2,4,6 and 7 (PF3D7_1417800, 1317100, 1355100, 0705400), replication factor C subunit 4 (PF3D7_1241700), chromatin assembly complex subunit CAF2 (PF3D7_1329300), and MutS homologue proteins MSH2,6 (PF3D7_1427500 and 0505500; Fig 3B, Tables EV5 and EV7). In *P. falciparum*, DNA helicase 60 was able to unwind double-stranded DNA in both directions and has been implicated in DNA replication (Pradhan *et al*, 2005). In other eukaryotic systems, MCMs 2–7 form a hexameric DNA helicase and are loaded onto origins of replication by the origin recognition complex to form the basis of a prereplication complex in G1 phase of the cell cycle (Ansari & Tuteja, 2012). The RFC complex is also involved in DNA replication initiation while MSH2 and MSH6

form a heterodimer to recognize DNA mismatches (Kunkel & Erie, 2005; Ohashi & Tsurimoto, 2017). CAF2 is a putative subunit of the CAF-1 chromatin assembly factor complex that assembles nucleosomes onto DNA during DNA replication and repair (Volk & Crispino, 2015).

The processes of DNA replication and repair have not been well characterized in *P. falciparum*, but as DNA replication has been reported to begin after 24 hpi (Matthews *et al*, 2018), it is unlikely that it would be taking place when the samples in this current study were collected (14 hpi). Protein components of the DNA replication and repair machinery were found to bind to the SPE2 DNA motif, but at a later stage of the IDC during DNA replication (Flueck *et al*, 2010). Thus, it is intriguing that these factors are associated with the *var* gene promoter during the G1 phase of the cell cycle.

### Identification of novel *var* gene-associated factors involved in chromatin structure

In addition to DNA replication and repair components, GO analysis showed that proteins that were significantly enriched in the promoter-targeted dCas9 immunoprecipitation compared to both the intron- and non-targeted dCas9 immunoprecipitations are significantly represented by the biological function categories of "chromatin assembly or disassembly" ($P = 1.77 \times 10^{-5}$) and "chromosome organization" ($P = 1.94 \times 10^{-5}$; Table EV10). These proteins include histone H2A.Z (PF3D7_0320900) and heterochromatin protein 1 (PF3D7_1220900), which have been shown to associate with the active and silent *var* gene promoters, respectively (Fig 3A and B) (Flueck *et al*, 2009; Petter *et al*, 2011). These data suggest that our method simultaneously purifies active and silent *var* gene promoters.

When compared to the intron-targeted dCas9 immunoprecipitation, the promoter-targeted dCas9 immunoprecipitation was enriched in several proteins involved in chromatin organization including a putative microchidia (MORC) family protein (PF3D7_1468100, ratio = 2.13, $P = 4.58 \times 10^{-11}$), a SWIB/MDM2 domain-containing protein (PF3D7_0518200, ratio = 1.91, $P = 9.67 \times 10^{-4}$), and three putative SNF2 domain-containing ATP-dependent chromatin remodelers: chromodomain-helicase-DNA-binding protein 1 ([CHD1, PF3D7_1023900] ratio = 2.8, $P =$

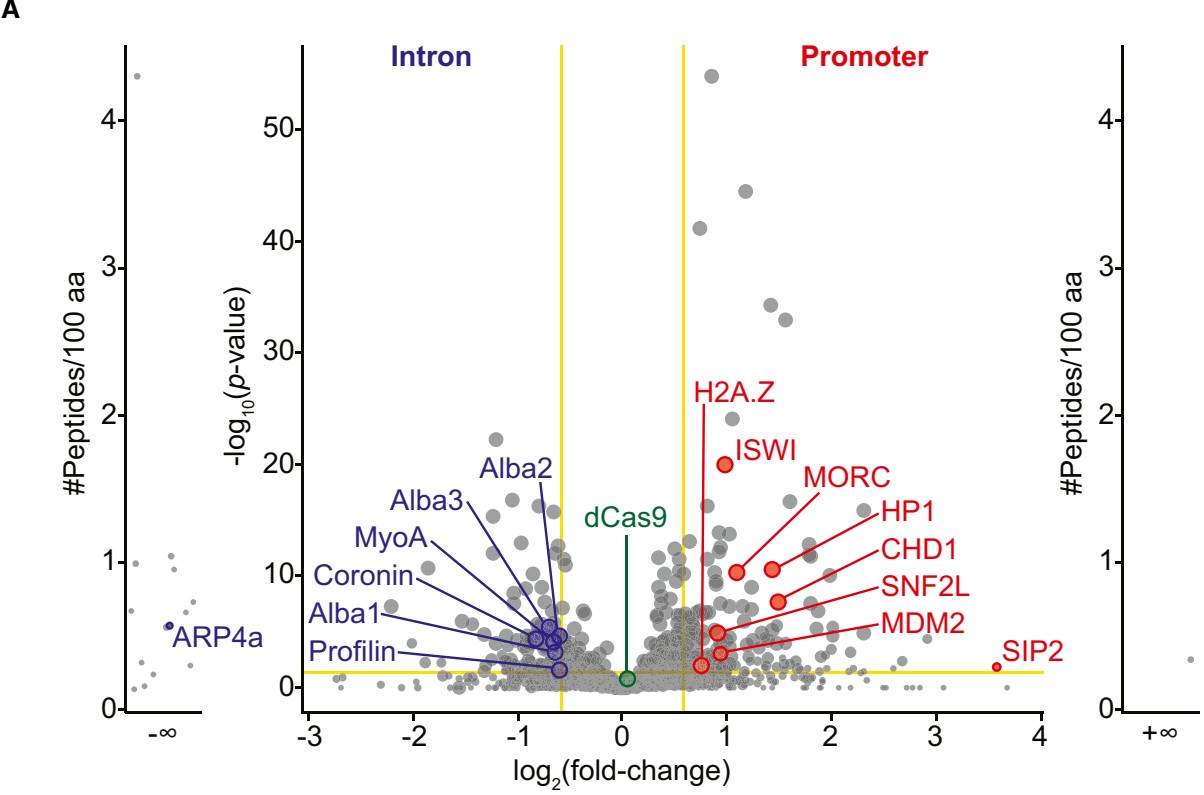

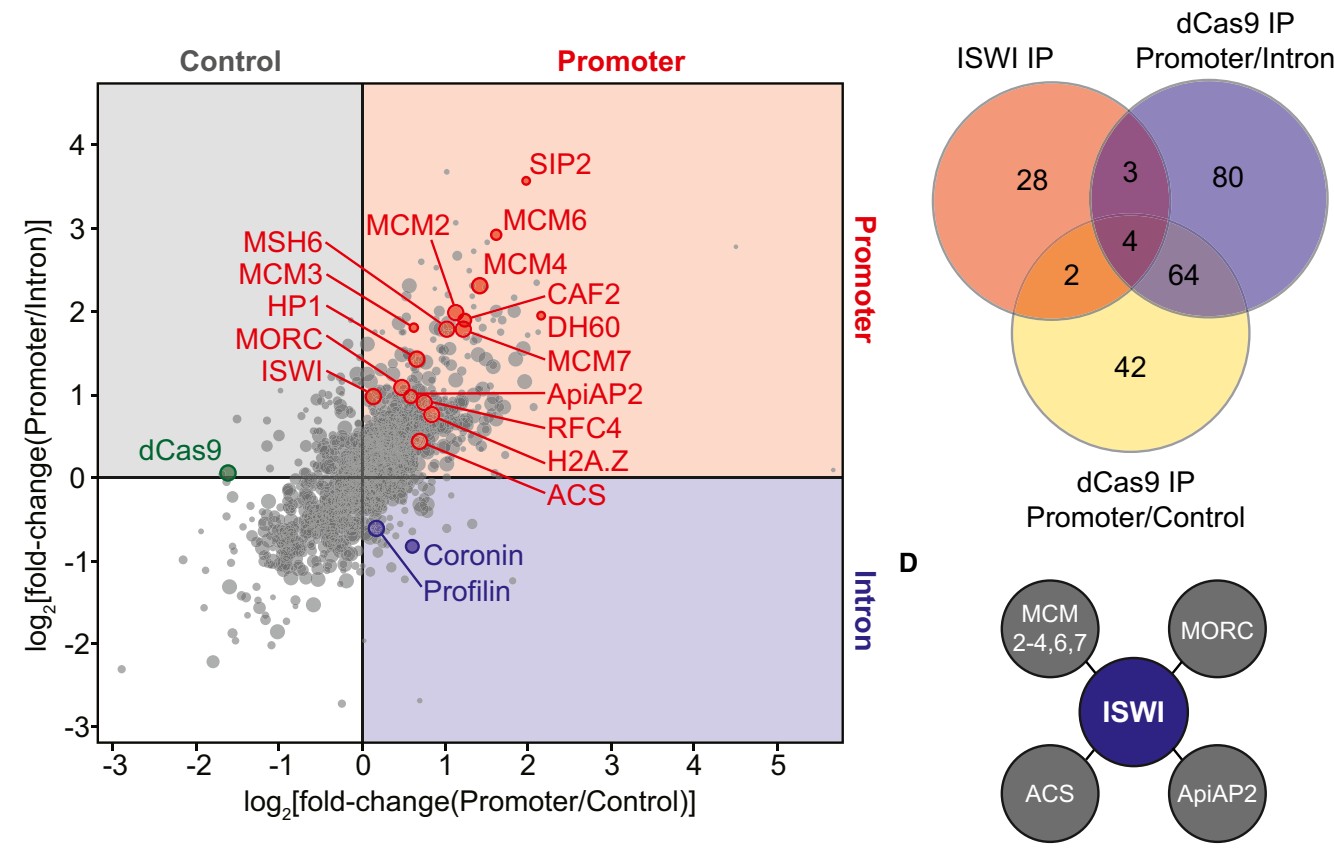

**Figure 3.**

**Figure 3. Proteomic analysis of dCas9-purified *var* gene introns and promoters.**

A  Volcano plot representation of label-free quantitative proteomic analysis of *P. falciparum* proteins present in intron- (left) and promoter-targeted (right) dCas9 immunoprecipitations. Each dot represents a protein, and its size corresponds to the sum of peptides from both conditions used to quantify the ratio of enrichment. For the main volcano plot, x-axis = $\log_2$(fold-change), y-axis = $-\log_{10}$(P-value), horizontal yellow line indicates adjusted P-value = 0.05, and vertical yellow lines indicate absolute fold-change = 1.5. Side panels indicate proteins uniquely identified in either sample (y-axis = number of peptides per 100 amino acids). dCas9 protein is highlighted in green for reference. Two replicates were used for the promoter-targeted dCas9, and four replicates were used for the intron-targeted dCas9. Fold enrichment and adjusted P-values for each protein highlighted in red (promoter) and blue (intron) can be found in Tables EV5 and EV6, respectively.

B  Correlation plot of label-free quantitative proteomic analysis of *P. falciparum* proteins enriched in promoter-targeted (top right quadrant in red) versus non-targeted control (top left quadrant in gray) or intron (bottom right quadrant in blue) dCas9 immunoprecipitations. Each dot represents a protein, and its size corresponds to the sum of peptides from both conditions used to quantify the ratio of enrichment. The x-axis = $\log_2$(fold-change[promoter/non-targeted control]) and y-axis = $\log_2$(fold-change[promoter/intron]). dCas9 protein is highlighted in green for reference. Two replicates were used for the promoter-targeted dCas9, and four replicates were used for the intron-targeted and non-targeted dCas9. Fold enrichment and adjusted P-values can be found for each protein highlighted in red (promoter) in Tables EV5 (intron comparison) and EV7 (control comparison) and for each protein highlighted in blue (intron) in Tables EV6 (promoter comparison) and EV8 (control comparison).

C  Venn diagram comparing proteins enriched in the ISWI IP LC-MS/MS (red) and the promoter-targeted dCas9 IP LC-MS/MS compared to the intron-targeted dCas9 (blue) or the non-targeted dCas9 (yellow).

D  Schematic of proteins shared between the ISWI IP LC-MS/MS and the promoter-targeted dCas9 IP LC-MS/MS (overlapping regions shown in C): ISWI (PF3D7_0624600), MORC (PF3D7_1468100), ApiAP2 (PF3D7_1107800), MCM2-4,6,7 (PF3D7_1417800, PF3D7_0527000, PF3D7_1317100, PF3D7_1355100, PF3D7_0705400), and acetyl-CoA synthetase (ACS, PF3D7_0627800).

$1.93 \times 10^{-8}$), SNF2L (PF3D7_1104200, ratio = 1.87, $P = 1.21 \times 10^{-5}$), and imitation switch ([ISWI, PF3D7_0624600] ratio = 1.97, $P = 1.02 \times 10^{-20}$; Fig 3A and Table EV5). In other eukaryotes, MORC proteins are able to topologically constrain DNA, which can lead to gene silencing via chromatin compaction (Koch *et al*, 2017; Kim *et al*, 2019). The SWIB domain can be found in proteins associated with the SWI/SNF chromatin-remodeling complex (Brownlee *et al*, 2015). In other eukaryotes, SWI/SNF, CHD1, and ISWI are chromatin-remodeling enzymes that use ATP hydrolysis to alter interactions between nucleosomes and DNA, leading to changes in chromatin organization/structure and accessibility to underlying DNA regulatory elements (reviewed in (Mueller-Planitz *et al*, 2013)). As such, these remodelers play important roles in chromosome structure, transcription regulation, and DNA replication, recombination, and repair (reviewed in (Erdel & Rippe, 2011; Narlikar *et al*, 2013)).

While the domain compositions of *Pf*CHD1 and *Pf*SNF2L are fairly conserved compared to those in other eukaryotes, the putative *Pf*ISWI and *Pf*MORC protein are more divergent (Fig EV3D and E). Comparison of the *Pf*ISWI amino acid sequence to those of other SNF2 domain-containing chromatin remodelers revealed a divergence in its SNF2 domain as well as the rest of the protein (Fig EV3D). *Pf*ISWI contains several zinc finger motifs and the characteristic N-terminal SNF2 ATPase and C-terminal helicase domains, but does not contain chromo-, bromo-, SANT, or SLIDE domains (Fig EV3D). *Pf*MORC contains a GHKL-type ATPase/kinase domain and coiled-coil domains similar to MORC proteins in other eukaryotes, but lacks the characteristic S5 fold domain believed to be important for its function in DNA compaction/loop formation (Fig EV3E) (Koch *et al*, 2017). However, *Pf*MORC does contain an N-terminal Kelch-type beta propeller, which seems to be unique to apicomplexan MORC proteins (Iyer *et al*, 2008).

Using CRISPR/Cas9, we attempted to epitope tag and/or knock out several of the most highly/significantly enriched factors at the *var* gene promoter such as ISWI, CHD1, MORC, MSH6, and CAF2 (Ghorbal *et al*, 2014). We were only successful in achieving an inducible knockdown system with ISWI, where sequences encoding a 3× hemagglutinin (3HA) tag followed by a *glmS*

ribozyme (Prommana *et al*, 2013) were inserted at the 3′ end of the corresponding gene, PF3D7_0624600. We performed immunoprecipitation followed by mass spectrometry of ISWI-3HA and found significant overlap (Fig EV3F) between ISWI-associated proteins and those enriched in the *var* gene promoter-targeted dCas9 IP LC-MS/MS analysis, including MCM2-4,6,7, acetyl-CoA synthetase (ACS, PF3D7_0627800), the putative MORC family protein, and an ApiAP2 protein (PF3D7_1107800) that was shown to strongly associate with *Pf*MORC in a recent protein interaction screen (Hillier *et al*, 2019) (Fig 3C and D; Tables EV5, EV7, EV11).

These results provide further experimental support for the efficacy of the CRISPR/dCas9 chromatin purification method. ISWI also associates with various transcriptional co-repressors and co-activators such as histone deacetylase 1 (HDAC1) and GCN5, respectively (Table EV11). These data suggest that ISWI may perform specific chromatin-related functions via different binding partners.

## ISWI is a transcriptional activator involved in *var* gene regulation

Because our dCas9 immunoprecipitation-proteomics approach did not distinguish between active and silent *var* gene promoters, functional characterization allowed us to unequivocally determine the role of ISWI in *var* gene regulation. As we were unable to achieve gene knockout, we used a gene knockdown (KD) approach, which allowed for the study of mutually exclusive expression in a clonal parasite population. Addition of glucosamine induced self-cleavage of the ribozyme and KD of ISWI at the protein level to an extent that did not affect growth and cell cycle progression (Figs 4A and EV4A).

A clone of ISWI-3HA-ribo was synchronized, glucosamine was added, and parasites were harvested 24 h later during ring stage, when *var* gene expression is at its highest. We performed RNA-seq and compared the data to the microarray time course data in Bozdech *et al* (2003) (Data ref: Bozdech *et al*, 2003) as in Lemieux *et al* (2009), which provided a statistical estimation of cell cycle progression at 12 hpi in both control and glucosamine-treated parasites (Fig EV4B). These data suggest that any differences in transcription were due to ISWI KD and not differences in cell cycle progression.

Indeed, glucosamine addition resulted in a significant knockdown (~ 50%, $q = 6.71 \times 10^{-10}$) of *iswi* transcript levels (Fig EV4C top, Table EV12) and a significant over-representation of down-regulated genes (562) compared to up-regulated genes (261; chi-square test $P < 0.0001$, Fig 4B, Table EV12). GO analysis showed that down-regulated genes are most significantly represented by the molecular function category of "structural constituent of ribosome" ($P = 2.82 \times 10^{-31}$, Table EV13), but there was no strong pattern of enrichment for up-regulated genes (Table EV14). Comparison of the RNA-seq data to the transcription time course in Bozdech *et al* (2003) (Data ref: Bozdech *et al*, 2003) revealed that the majority of significantly down-regulated genes normally reach their peak transcript level in ring stage parasites between 8 and 18 hpi; however, the same trend is not seen in significantly up-regulated genes (Fig 4C).

To investigate the genome-wide occupancy of ISWI-3HA, we performed ChIP-seq in ring stage parasites. These data showed ISWI enrichment in the promoter regions of genes that were down-regulated upon ISWI KD (Fig 4D). When compared to published Assay for Transposase-Accessible Chromatin using sequencing (ATAC-seq) data from a similar time point in the IDC (Toenhake *et al*, 2018; Data ref: Toenhake *et al*, 2018), our ChIP-seq data revealed an enrichment of ISWI that overlapped with accessible chromatin in intergenic regions (Fig 4E). Indeed, if all *P. falciparum* genes were divided into quartiles based on their transcript levels at 12 hpi, ISWI was most enriched in the promoter regions of genes with the highest transcript levels (Fig 4F). These data suggest that ISWI

binding to promoter regions contributes to transcriptional activation in real time.

Among the genes that were significantly down-regulated upon ISWI KD was the *var* gene that is active in this clone (PF3D7_1240600; $q = 2.53 \times 10^{-5}$, Figs 4B and EV4C bottom, Table EV12). However, mutually exclusive expression of the *var* gene family was maintained during ISWI KD (Fig EV4C bottom). Interestingly, ISWI and H3K9ac ChIP-seq showed enrichment of both these features in the promoter region of the active *var* gene (Fig 4G and H). In comparison, ISWI and H3K9ac were largely absent from the promoters of the silent *var* genes (Figs 4H and EV4D). Taken together, these data suggest that ISWI plays a direct role in *var* gene activation, but not repression, in ring stage parasites.

## Discussion

CRISPR/Cas9 continues to transform genetics in apicomplexan parasites, but the full range of CRISPR-based tools has yet to be developed in *Plasmodium*. dCas9 has been used in mammalian systems to purify specific loci in the genome along with closely associated proteins, DNA, and RNA (Fujita & Fujii, 2013; Waldrip *et al*, 2014; Liu *et al*, 2017; Myers *et al*, 2018). Importantly, this method is able to capture chromatin complexes that may not be detected with *in vitro* affinity purification assays. In this study, we purified specific chromatin loci without the use of genome editing, which allowed us

---

**Figure 4.  ISWI is a transcriptional activator involved in *var* gene regulation.**

A   Western blot analysis of cytoplasmic (Cty) and nuclear extracts from a bulk population of ISWI-3HA-ribo parasites in the absence (−) or presence (+) of glucosamine (GlcN). ISWI-3HA is detected with an anti-HA antibody. Antibodies against aldolase (Ald.) and histone H3 are controls for the cytoplasmic and nuclear extracts, respectively. Molecular weights are shown to the right.

B   MA plot of $\log_2$(glucosamine-treated/untreated, M) plotted over the mean abundance of each gene (A) at 12 hpi. Transcripts with a significantly higher (above *x*-axis) or lower (below *x*-axis) abundance in the presence of glucosamine are highlighted in red ($q \leq 0.05$). *iswi* is highlighted in blue ($q = 6.71 \times 10^{-10}$), and the active *var* gene is highlighted in green ($q = 2.53 \times 10^{-5}$). Two and three replicates were used for untreated and glucosamine-treated parasites, respectively. *P*-values were calculated with a Wald test for significance of coefficients in a negative binomial generalized linear model as implemented in DESeq2 (Love *et al*, 2014). $q = B$onferroni corrected *P*-value.

C   Frequency plot showing the time in the IDC of peak transcript level (comparison to microarray time course in Bozdech *et al* (2003) (Data ref: Bozdech *et al*, 2003)) for genes that are significantly down-regulated (blue) or up-regulated (yellow) following ISWI knockdown.

D   Meta-gene plot showing average ISWI enrichment (*y*-axis = ChIP/Input) in clonal ISWI-3HA parasites at 12 hpi from 1.5 kb upstream of the translation start site ("Start") to 1.5 kb downstream of the translation stop site ("Stop") for genes that are down- (blue) or up-regulated (yellow) upon ISWI knockdown. One replicate was used for the ISWI ChIP-seq.

E   ChIP-seq data show enrichment of ISWI (blue) in clonal ISWI-3HA parasites at 12 hpi relative to regions of accessible chromatin ("ATAC," red). Chromatin accessibility (ATAC-seq) data were taken from the 15 hpi time point in Toenhake *et al*, 2018 (Data ref: Toenhake *et al*, 2018). Genome location is indicated at the top of the panel. The *x*-axis is DNA sequence, with genes represented by black boxes indented to delineate introns and labeled with white arrowheads to indicate transcription direction. The *y*-axis is enrichment (ChIP/Input or ATAC-seq/genomic DNA). One replicate was used for the ISWI ChIP-seq.

F   Meta-gene plot showing average ISWI enrichment (*y*-axis = ChIP/Input) in clonal ISWI-3HA parasites at 12 hpi from 1.5 kb upstream of the translation start site ("Start") to 1.5 kb downstream of the translation stop site ("Stop") for all genes, which are grouped into quartiles based on their transcript levels (RPKM) at 12 hpi. Dark red represents genes with the highest transcript levels ("1st"), light red represents genes with the second highest transcript levels ("2nd"), orange represents genes with the third highest transcript levels ("3rd"), and yellow represents genes with the lowest transcript levels ("4th"). One replicate was used for the ISWI ChIP-seq, and the transcription data are an average from the two replicates of the untreated ISWI-3HA clone used for the differential expression analysis.

G   ChIP-seq data show enrichment of ISWI (blue) and H3K9ac (green) in clonal ISWI-3HA parasites at 12 hpi at the active *var* gene (PF3D7_1240600). RNA-seq data from this clone at 12 hpi show transcript levels for this gene (gray). Genome location is indicated at the top of the panel. The *x*-axis is DNA sequence, with genes represented by black boxes indented to delineate introns and labeled with white arrowheads to indicate transcription direction. The *y*-axis is ChIP/Input for ChIP data and RPKM for RNA-seq data. One replicate was used for each ChIP-seq, and the RNA-seq data are from a single replicate from the untreated ISWI-3HA clone used for the differential expression analysis.

H   Meta-gene plot showing average ISWI enrichment (*y*-axis = ChIP/Input) in clonal ISWI-3HA parasites at 12 hpi from 1.5 kb upstream of the translation start site ("Start") to 1.5 kb downstream of the translation stop site ("Stop") for the active *var* gene (PF3D7_1240600, blue) or silent *var* genes (yellow). One replicate was used for the ISWI ChIP-seq.

Source data are available online for this figure.

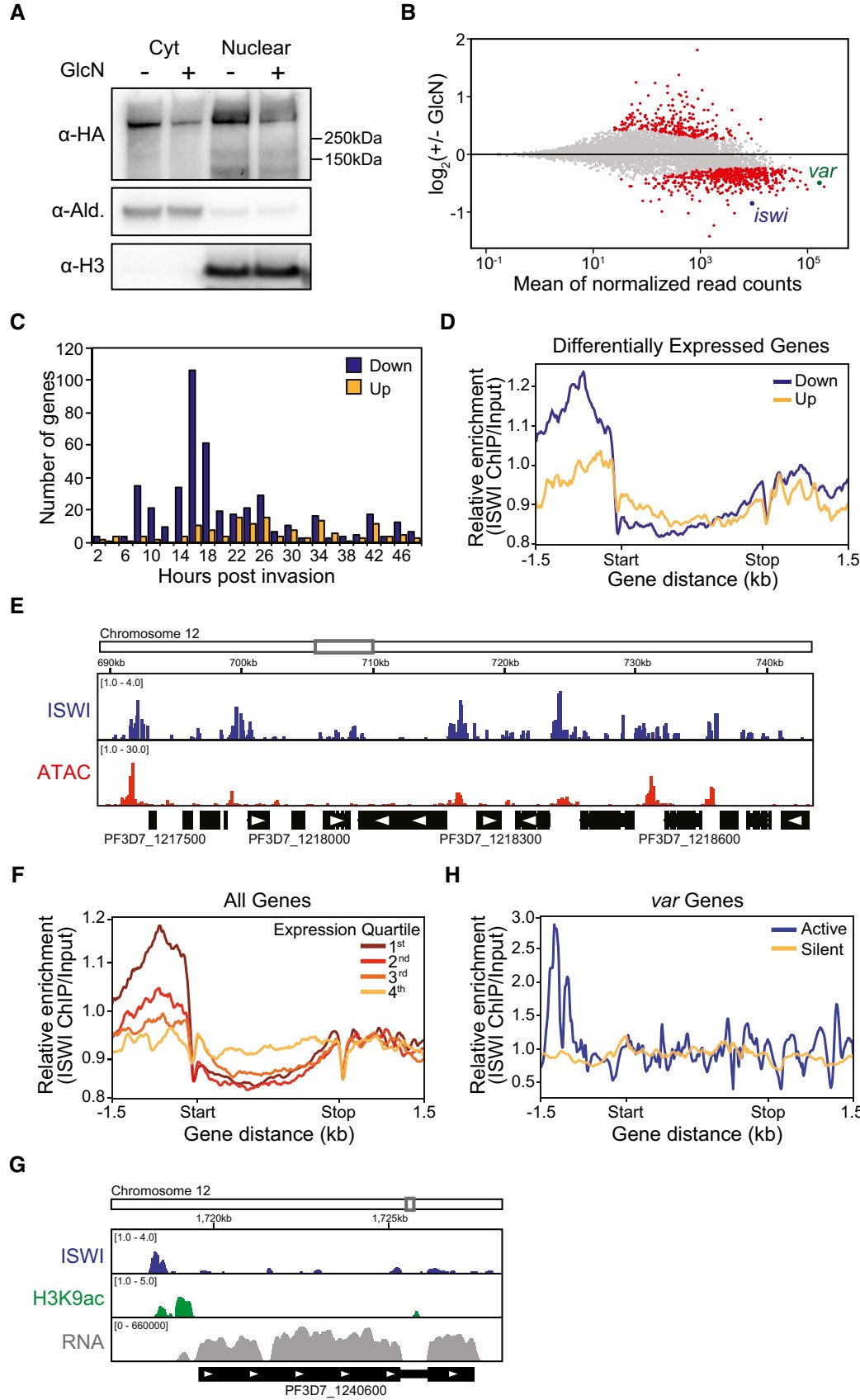

Figure 4.

to identify factors bound in a biologically relevant spatio-temporal context.

Traditionally, single-locus dCas9 immunoprecipitation for proteomic analysis has proven challenging even in model systems due to the high amount of input material needed to overcome non-specific background. The future development of an inducible, biotin-based CRISPR/dCas9 IP LC-MS/MS analysis system (Liu *et al*, 2017; Myers *et al*, 2018) in *Plasmodium* may allow for single-locus purification. Many seminal studies attempting to purify a specific genomic sequence maximized the immunoprecipitated protein content per cell by targeting telomeric repeats, which are present in high numbers within the genome (Déjardin & Kingston, 2009; Fujita *et al*, 2013; Liu *et al*, 2017). For our analysis, we exploited the homology of targeted sequences within the *var* gene family and maximized protein content of the immunoprecipitation by targeting dCas9 to multiple *var* genes in the same strain with a single guide RNA (Figs 1 and EV1A and C). We used ChIP-seq to demonstrate that dCas9 bound robustly to *var* gene regulatory elements (Fig 2). It is possible that we immunoprecipitated both silent and active *var* genes simultaneously from non-clonal bulk cultured parasites, as they collectively did not express a single *var* gene (Fig EV2D). Importantly, RNA-seq and dCas9 RIP-seq of the *var* promoter-targeted strains showed that dCas9-targeted *var* genes are transcribed and that mutually exclusive expression is maintained (Figs 2C and EV2B–E). Thus, this technology could be used for investigation of other multigene families in *P. falciparum* and other parasites.

Label-free quantitative proteomics and comparison of the promoter- and intron-targeted dCas9 immunoprecipitations to each other and to the non-targeted dCas9 immunoprecipitation allowed for the identification of proteins that were specific to each DNA element (Fig 3). Using a non-targeted dCas9 that does not bind specifically to the genome for normalization, we identified *var* gene-bound factors such as SIP2 and other proteins that were shown to bind to or near *var* genes like the Alba proteins, HP1, and H2A.Z (Flueck *et al*, 2009; Pérez-Toledo *et al*, 2009; Petter *et al*, 2011; Chêne *et al*, 2012; Goyal *et al*, 2012). However, our analysis did not identify as significantly enriched several chromatin-associated proteins that have been implicated in mutually exclusive *var* gene transcription including HDA2, SIR2, SET2, SET10, and RECQ1 (Freitas-Junior *et al*, 2005; Tonkin *et al*, 2009; Volz *et al*, 2012; Jiang *et al*, 2013; Coleman *et al*, 2014; Ukaegbu *et al*, 2014; Li *et al*, 2019). SIR2, SET2, SET10, and RECQ1 were not detected in any sample, perhaps due to very low abundance or to weak/transient binding to *var* genes (Source Data for Tables EV5–EV8). It is also possible that these factors associate with *var* genes before or after the 14 hpi time point we analyzed. Indeed, SET10 was shown by IFA to associate with the active *var* gene at later stages, perhaps to facilitate gene poising (Volz *et al*, 2012).

Importantly, we identified novel *var* gene-associated factors that have thus far not been implicated in *var* gene regulation, but provide important insight into *var* gene biology. The *var* gene promoter was enriched in members of the DNA prereplication complex. It is possible that the targeted *var* gene promoters are located near origins of replication where the prereplication complex is loaded during G1 phase (ring stage parasites), as in other eukaryotes (Ansari & Tuteja, 2012). However, the presence of DNA mismatch repair (MMR) components suggests that CAF2 and the MCM complex might be involved in DNA damage repair. Indeed, recent evidence showed that *var* genes frequently recombine during asexual replication, leading to crossover between different *var* sequences in a manner that preserves the reading frame (Claessens *et al*, 2014). These recombination events are often located near putative G-quadruplex DNA motifs, which form secondary structures that can interfere with transcription or lead to DNA recombination by stalling DNA polymerase (Siddiqui-Jain *et al*, 2002; Koole *et al*, 2014; Stanton *et al*, 2016). In *P. falciparum*, half of the total predicted G-quadruplex motifs are within or upstream of *var* genes, especially of the *upsB* type, which we target with dCas9 in this study (Smargiasso *et al*, 2009). Thus, the MMR machinery, MCM helicases, and CAF-1 complex could bind near *var* gene promoters to help maintain chromatin stability and DNA repair in the recombinogenic *var* gene family during asexual replication.

The *var* gene promoters were also enriched in chromatin remodelers and MORC, which indicates the importance of chromatin structure at this regulatory element. It has been demonstrated that chromatin accessibility and incorporation of histone variants H2A.Z and H2B.Z at *P. falciparum* gene promoters correlate with transcriptional activity, but the factors behind these phenomena are unknown (Hoeijmakers *et al*, 2013; Petter *et al*, 2013; Ruiz *et al*, 2018; Toenhake *et al*, 2018). Thus, our functional characterization of ISWI provides novel insight into transcriptional activation in *P. falciparum*. We showed that ISWI binds to the promoters of active genes and intergenic regions of high chromatin accessibility in ring stage parasites (Fig 4D–F). Although we showed that ISWI associates with proteins involved in transcriptional repression and activation, there was a significant over-representation of down-regulated genes upon ISWI knockdown, the majority of which normally reach peak transcription level in ring stage parasites (Fig 4B and C). One of these down-regulated genes was the active *var* gene, which had an enrichment of ISWI at its promoter (Fig 4G and H). These data suggest that ISWI binds to upstream regions of genes to facilitate transcriptional activation, perhaps through changes in chromatin composition at the *var* gene promoter.

It is possible that ISWI achieves specificity and modulates nucleosome positioning/composition either directly or via binding partners, as has been shown for many chromatin remodelers in other eukaryotic systems (reviewed in (Erdel & Rippe, 2011)). Indeed, we show that the structurally divergent *Pf*ISWI contains zinc finger domains and interacts with diverse proteins such as MORC, an ApiAP2 transcription factor, HDAC1, and ACS (Fig 3D). Interestingly, recent studies have demonstrated that enzymes involved in acetyl-CoA synthesis localize to specific genomic loci to effect changes in histone acetylation, which can enhance gene transcription or DNA repair (reviewed in (Li *et al*, 2018)). Likewise, localization of *Pf*ACS to the promoters of genes could have important implications for the relationship between gene expression and metabolism.

As the *Pf*MORC sequence diverges from that of MORC proteins found in other eukaryotes, it is unclear what transcriptional role it might play with ISWI in *P. falciparum*. A recent study in another apicomplexan parasite, *Toxoplasma gondii*, showed that MORC associates with ApiAP2 transcription factors and a histone deacetylase (HDAC3) to transcriptionally silence specific cohorts of genes important for sexual differentiation (Farhat *et al*, 2020). This MORC/HDAC3 complex seems to modulate nucleosome positioning

and/or histone acetylation to directly silence transcription or act as a barrier to neighboring euchromatin. A similar model could prove true in *P. falciparum*, where an ApiAP2 transcription factor (such as PF3D7_1107800) recruits ISWI and other cofactors such as MORC, HDAC1, or GCN5 to alter the local chromatin structure, which could lead to gene activation, repression, or insulation from neighboring HP1-mediated heterochromatin. Future functional characterization of the *Pf*ISWI complex will elucidate specific roles in chromatin composition and transcription, especially with regard to *var* genes.

In addition, many proteins of unknown function without homologs in other organisms were enriched in our dCas9 IP LC-MS/MS analysis. Because *P. falciparum* lacks CTCF and many conserved transcription factors found in other eukaryotes, the proteins identified in this study may prove to be missing links in *var* gene regulation and higher-order chromatin organization (Aravind *et al*, 2003; Coulson *et al*, 2004; Templeton *et al*, 2004). Future functional characterization of these factors will reveal their roles in chromatin organization and unique transcriptional regulation of the *var* gene family.

# Materials and Methods

## Reagents and Tools table

| Reagent/Resource | Reference or Source | Identifier or Catalog Number |
|---|---|---|
| **Experimental Models** | | |
| *3D7 strain (P. falciparum)* | (Walliker *et al*, 1987) PMID 3299700 | |
| 3D7 ISWI-3HA-ribo (*P. falciparum*) | This study | |
| 3D7 pUF1-dCas9-3HA, pL7_*var*promoter_IP | This study | |
| 3D7 pUF1-dCas9-3HA, pL7_*var*intron_IP | This study | |
| 3D7 pUF1-dCas9-3HA, pL7_Control_IP | This study | |
| **Recombinant DNA** | | |
| pL7_0624600_A | This study | |
| pL7_0624600_B | This study | |
| pL7_Control_IP | This study | |
| pL7_*var*intron_IP | This study | |
| pL7_*var*promoter_IP | This study | |
| pUF1-Cas9 | (Ghorbal *et al*, 2014) PMID 24880488 | |
| pUF1-dCas9-3HA | This study | |
| **Antibodies** | | |
| HRP-conjugated donkey anti-rabbit (1:5,000 for western blot) | Sigma | Cat # GENA934 |
| Rabbit IgG | Diagenode | Cat # C15410206 |
| Rabbit polyclonal anti-H3 (1:1,000 for western blot) | Abcam | Cat # ab1791 |
| Rabbit polyclonal anti-H3K9ac (1 µg/25uL Dynabeads for immunoprecipitation) | Millipore | Cat # 07-352 |
| Rabbit polyclonal anti-HA (1:1,000 for western blot, 1 µg/25 µl Dynabeads for immunoprecipitation) | Abcam | Cat # ab9110 |
| Rabbit polyclonal HRP-conjugated anti-aldolase (1:5,000 for western blot) | Abcam | Cat # ab38905 |
| **Oligonucleotides and other sequence-7based reagents** | | Sequence (5′–3′) |
| pL7_0624600_A sgRNA | This study | ATCATTATTACTTCTGTCTG |
| pL7_0624600_B sgRNA | This study | ATGTCGTTTTAATTAATTTC |
| pL7_*var*intron_IP sgRNA | This study | GTAAATGTGTATGTAGTTAT |
| pL7_*var*promoter_IP  sgRNA | This study | GGTTTATATCGTGGCATACA |

                                                    

**Reagent and Tools table**  (continued)

| Reagent/Resource | Reference or Source | Identifier or Catalog Number |
|---|---|---|
| **Chemicals, Enzymes and other reagents** | | |
| 10% Albumax | Thermo Fisher | Cat # 11020 |
| 4–12% Bis-Tris NuPage gel | Thermo Fisher | Cat # NP0321 |
| Anti-HA Dynabeads | Thermo Fisher | Cat # 88836 |
| Benzonase | Merck | Cat # 71206 |
| C18 column (50 cm × 75 μm; nanoViper Acclaim PepMap™ RSLC, 2 μm, 100 Å) | Thermo Fisher | Cat # 164535 |
| C18 column (75 μm inner diameter × 2 cm; nanoViper Acclaim PepMap™ 100) | Thermo Fisher | Cat # 164942 |
| C18 StageTips | In house | |
| C18 ziptips (for ISWI-HA IP/MS) | Merck | Cat # ZTC04S096 |
| D-(+)-Glucosamine hydrochloride | Sigma | Cat # G1514 |
| Dithiobissuccinimidyl propionate | Thermo Fisher | Cat # 22585 |
| Dithiothreitol | Sigma | Cat # D9779 |
| DNase Turbo | Thermo Fisher | Cat # AM2238 |
| DSM1 | MR4/BEI Resources | Cat # MRA-1161 |
| Dulbecco's phosphate-buffered saline | Thermo Fisher | Cat # 14190 |
| EASY-Spray column, 50 cm × 75 μm ID, PepMap RSLC C18, 2 μm | Thermo Fisher | Cat # ES803A |
| Gentamicin | Sigma | Cat # G1397 |
| Glycine | Sigma | Cat # G8898 |
| Glycogen | Thermo Fisher | Cat # 10814 |
| Hypoxanthine | C.C.Pro | Cat # Z-41-M |
| In-Fusion HD Cloning Kit | Clontech | Cat # 639649 |
| Iodoacetamide | Sigma | Cat # I1149 |
| KAPA HiFi DNA Polymerase | Roche | Cat # 07958846001 |
| Methanol-free formaldehyde | Thermo Fisher | Cat # 28908 |
| $NH_4HCO_3$ | Sigma | Cat # 09830 |
| NuPage Reducing Agent | Thermo Fisher | Cat # NP0004 |
| NuPage Sample Buffer | Thermo Fisher | Cat # NP0008 |
| Plasmion | Fresenius Kabi | |
| Protease Inhibitor Cocktail | Roche | Cat # 11836170001 |
| Protein G Dynabeads | Invitrogen | Cat # 10004D |
| Proteinase K | New England Biolabs | Cat # P8107S |
| RNaseA | Thermo Fisher | Cat # EN0531 |
| RNasin RNase inhibitor | Promega | Cat # N2511 |
| RPMI-1640 culture media | Thermo Fisher | Cat # 11875 |
| Saponin | Sigma | Cat # S7900 |
| Sorbitol | Sigma | Cat # S6021 |
| SuperSignal West Pico chemiluminescent substrate | Thermo Fisher | Cat # 34080 |
| SYBR Green I | Sigma | Cat # S9430 |
| Trizol LS | Thermo Fisher | Cat # 10296010 |
| Trypsin (for ISWI-HA IP/MS) | Thermo Fisher | Cat # 90059 |
| Trypsin/LysC | Promega | Cat # V5071 |
| WR99210 | Jacobus Pharmaceuticals | |
| XL10-Gold Ultracompetent *Escherichia coli* | Agilent Technologies | Cat # 200315 |

**Reagent and Tools table** (continued)

| Reagent/Resource | Reference or Source | Identifier or Catalog Number |
|---|---|---|
| Software | | |
| FigTree (v 1.4.3) | http://tree.bio.ed.ac.uk/software/figtree/ | |
| GraphPad Prism | www.graphpad.com | |
| Integrative Genomics Viewer | (Robinson et al, 2011) | |
| Mascot (v 2.4.1) | (Perkins et al, 1999) | |
| myProMS (v 3.9) | (Poullet et al, 2007) | |
| Proteome Discoverer | Thermo Fisher | |
| Other | | |
| 2100 Bioanalyzer | Agilent | |
| Bioruptor Pico | Diagenode | Cat # B01060010 |
| ChemiDoc XRS+ | Bio-Rad | Cat # 1708265 |
| Dynabeads mRNA DIRECT Kit | Thermo Fisher | Cat # 61012 |
| DynaMag | Thermo Fisher | Cat # 12321D |
| Guava easyCyte Flow Cytometer | Merck Millipore | |
| HPLC | Thermo Fisher | Cat # Easy-nLC1000 |
| MicroPlex library preparation kit (v2) | Diagenode | Cat # C05010014 |
| miRNeasy Mini Kit | Qiagen | Cat # 217004 |
| NextSeq 500 | Illumina | Cat # SY-415-1002 |
| Q Exactive HF-X mass spectrometer | Thermo Fisher | |
| RNeasy MinElute Cleanup Kit | Qiagen | Cat # 74204 |
| Sonication tubes | Diagenode | Cat # C30010016 |
| TruSeq stranded mRNA LT library prep kit | Illumina | Cat # 20020594 |

## Methods and Protocols

### Parasite culture

Asexual blood stage 3D7 *P. falciparum* parasites were cultured as previously described in (Lopez-Rubio *et al*, 2009). Briefly, parasites were cultured in human RBCs (obtained from the Etablissement Français du Sang with approval number HS 2016-24803) in RPMI-1640 culture medium (Thermo Fisher 11875) supplemented with 10% v/v Albumax I (Thermo Fisher 11020), hypoxanthine (0.1 mM final concentration, C.C.Pro Z-41-M) and 10 mg gentamicin (Sigma G1397) at 4% hematocrit and under 5% $O_2$, 3% $CO_2$ at 37°C. Parasite development was monitored by Giemsa staining. Parasites were synchronized by sorbitol (5%, Sigma S6021) lysis at ring stage, plasmagel (Plasmion, Fresenius Kabi) enrichment of late stages 24 h later, and an additional sorbitol lysis 6 h after plasmagel enrichment. The 0 h time point was considered to be 3 h after plasmagel enrichment. Parasites were harvested at 1–5% parasitemia.

### Generation of strains

All cloning was performed using KAPA HiFi DNA Polymerase (Roche 07958846001), In-Fusion HD Cloning Kit (Clontech 639649), and XL10-Gold Ultracompetent *E. coli* (Agilent Technologies 200315). Transgenic dCas9 parasites were generated using a two-plasmid system (pUF1 and pL7) based on the CRISPR/Cas9 system previously described in (Ghorbal *et al*, 2014). The sequence encoding the 3xFLAG-Cas9 in the pUF1 plasmid was

replaced with a gene encoding a catalytically inactive Cas9 (with RuvC and HNH mutations developed in (Qi *et al*, 2013)), which was fused at its carboxy terminus with a 3× hemagglutinin (HA) tag. The pL7 sgRNA expression plasmid was modified to encode an optimized sgRNA (with duplex extension and thymine mutation as developed in (Dang *et al*, 2015)). For pL7_*var*Promoter_IP, the non-coding strand sequence 5′-GGTTTATATCGTGGCATACA-3′ approximately 40 bp upstream of the translation start site of 19 *var* genes was ordered from Eurofins Genomics (Ebersberg, Germany) and inserted into the optimized pL7 sgRNA expression plasmid. For pL7_*var*Intron_IP, the non-coding strand sequence 5′-GTAAATGTGTATGTAGTTAT-3′ in region III of the intron (as defined in (Calderwood *et al*, 2003)) of 13 *var* genes was ordered from Eurofins Genomics (Ebersberg, Germany) and inserted into the optimized pL7 sgRNA expression plasmid. The pL7_Control_IP plasmid contains the empty cloning site 5′-CCTAGGAACT-CATCGCTCGCGATGCTGCCCGACA-3′ as the sgRNA. Short read alignment using bowtie analysis shows that neither this sequence nor the final 20 nucleotide (nt) sequence adjacent to the tracrRNA correspond to any sequence in the *P. falciparum* 3D7 genome with three or less mismatches (Fig EV2A). A 3D7 wild-type bulk ring stage culture was transfected with 25 μg each of pUF1 and pL7, and continuous drug selection was applied with 1.33 nM WR99210 (Jacobus Pharmaceuticals) and 0.75 μM DSM1 (MR4/BEI Resources MRA-1161).

The ISWI-3HA-ribo strain was generated with CRISPR/Cas9 as in (Ghorbal *et al*, 2014) by transfecting a 3D7 wild-type bulk ring stage culture with 25 μg pUF1-Cas9 and 12.5 μg each of two pL7 plasmids containing sgRNA-encoding sequences 5′-ATCATTATTACTTC TGTCTG-3′ and 5′-ATGTCGTTTTAATTAATTTC-3′ (ordered from Eurofins Genomics [Ebersberg, Germany]) targeting the 3′ end of the PF3D7_0624600 coding sequence. Each pL7 plasmid contained the homology repair construct 5′-ATGATAAGAGCAA AAATGTGAGGAATGATGACGAAGATGACGATGATGATGAAGATGA TGATGAAGAGGATGACGAAGATAAAAATGAAAGTTCAAATTATAA TAATAATAAGAAAAAAAAGACAAATACTTCAAGTAGAAATAGCAG TAATAATAATAGTAGTAATAAAAATAAAAATAATAAAAGTGGTAA TGATATTCATCAAGCTAGTAACTTACTGTATCAAAATTTATTAAAT AATCCGCAAAGTCTTTTACAGCATTTAAATTTGGAAGATGTAAAGA ATTTCTTAAAAGCTGCTGACAGAAGTAATAATGATAACTTACCAG AAATTAATGGCGGTGGATACCCTTACGATGTGCCTGATTACGCGTA TCCCTATGACGTACCAGACTATGCGTACCCTTATGACGTTCCGGAT TATGCTCACGGGGTGTAAGCGGCCGCGGTCTTGTTCTTATTTTCTC AATAGGAAAAGAAGACGGGATTATTGCTTTACCTATAATTATAGC GCCCGAACTAAGCGCCCGGAAAAAGGCTTAGTTGACGAGGATGGA GGTTATCGAATTTTCGGCGGATGCCTCCCGGCTGAGTGTGCAGAT CACAGCCGTAAGGATTTCTTCAAACCAAGGGGGTGACTCCTTGAA CAAAGAGAAATCACATGATCTTCCAAAAAACATGTAGGAGGGGA CAACGACATTTCTTTAATAAATAAATTAATATATATATATATATA TATAGAGAGAGAGAGAAATATTATATTTGATATATGTAGCTGC GAATGTTTAATTTTTTAGTTTATATATTTTGGAAATGTGTCTTTTG AATTTTTTTTTATTTTGAAAAGATGAATGATATGATATATGATA TATGAGGATATAATAAGAAGGAAATATATCTATATGTATATATAT ATATATATATGTATATTTTAGTTTTTGAGAGAATTTTATTTTCAGA AGGATATAAAAAAGAAAAAAATGGATAA-3′ (synthesized by GenScript Biotech [Piscataway, NJ, USA]). This repair construct contains a 3xHA-encoding sequence followed by a *glmS* ribozyme sequence (Prommana *et al*, 2013), which are flanked by 300 bp homology repair regions upstream and downstream of the Cas9 cut sites. Two silent shield mutations were made in the homology repair region to prevent further Cas9 cutting. After transfection, drug selection was applied for 7 days at 2.67 nM WR99210 (Jacobus Pharmaceuticals) and 1.5 μM DSM1. Parasites reappeared ~ 3 weeks after transfection, and 5-fluorocytosine was used to negatively select the pL7 plasmid. Parasites were cloned by limiting dilution, and the targeted genomic locus was sequenced to confirm tag and ribozyme integration.

### Western blot analysis

For the Western blot in Fig 1C, extracts prepared during the dCas9 immunoprecipitation protocol (see below) for LC-MS/MS were supplemented with NuPage Sample Buffer (Thermo Fisher NP0008) and NuPage Reducing Agent (Thermo Fisher NP0004) and denatured for 5 min at 95°C. For Fig 4A, 2 ml of iRBC (3% parasitemia) was synchronized and split into two 1 ml cultures at 3 hpi, and glucosamine (1.25 mM, Sigma G1514) was added to one culture. After 24 h, iRBCs were washed once with Dulbecco's phosphate-buffered saline (DPBS, Thermo Fisher 14190) at 37°C and lysed with 0.075% saponin (Sigma S7900) in DPBS at 37°C. Parasites were washed once with DPBS, resuspended in 1 ml cytoplasmic lysis buffer (25 mM Tris–HCl pH 7.5, 10 mM NaCl, 1.5 mM MgCl$_2$, 1% IGEPAL CA-630, and 1× protease inhibitor cocktail ["PI", Roche 11836170001]) at 4°C, and incubated on ice for 30 min. Cells were

further homogenized with a chilled glass douncer, and the cytoplasmic lysate was cleared with centrifugation (13,500 *g*, 10 min, 4°C). The pellet (containing the nuclei) was resuspended in 100 μl nuclear extraction buffer (25 mM Tris–HCl pH 7.5, 600 mM NaCl, 1.5 mM MgCl$_2$, 1% IGEPAL CA-630, PI) at 4°C and sonicated for 10 cycles with 30 s (on/off) intervals (5 min total sonication time) in a Diagenode Pico Bioruptor at 4°C. This nuclear lysate was cleared with centrifugation (13,500 *g*, 10 min, 4°C). Protein samples were supplemented with NuPage Sample Buffer (Thermo Fisher NP0008) and NuPage Reducing Agent (Thermo Fisher NP0004) and denatured for 10 min at 70°C. Proteins were separated on a 4–12% Bis-Tris NuPage gel (Thermo Fisher NP0321) and transferred to a PVDF membrane. The membrane was blocked for 1 h with 1% milk in PBST (PBS, 0.1% Tween 20) at 25°C. HA-tagged proteins and histone H3 were detected with anti-HA (Abcam ab9110, 1:1,000 in 1% milk-PBST) and anti-H3 (Abcam ab1791, 1:1,000 in 1% milk-PBST) primary antibodies, respectively, followed by donkey anti-rabbit secondary antibody conjugated to horseradish peroxidase ("HRP", Sigma GENA934, 1:5,000 in 1% milk-PBST). Aldolase was detected with anti-aldolase-HRP (Abcam ab38905, 1:5,000 in 1% milk-PBST). HRP signal was developed with SuperSignal West Pico chemiluminescent substrate (Thermo Fisher 34080) and imaged with a ChemiDoc XRS+ (Bio-Rad).

### dCas9 chromatin immunoprecipitation

1   Using a DynaMag magnet (Thermo Fisher 12321D), wash 25 μl Protein G Dynabeads (Invitrogen 10004D) twice with 1 ml ChIP dilution buffer (16.7 mM Tris–HCl pH 8, 150 mM NaCl, 1.2 mM EDTA pH 8, 1% Triton X-100, 0.01% SDS).

2   Resuspend beads in 1 ml ChIP dilution buffer and add 1 μg of anti-HA antibody (Abcam ab9110). Incubate on a rotator at 4°C for ~ 6 h.

3   Cross-link synchronized parasites (~ 2 × 10$^8$ parasites) by adding methanol-free formaldehyde (Thermo Fisher 28908) to the culture (final concentration 1%) and incubating with gentle agitation for 10 min at 25°C.

4   Quench the cross-linking reaction by adding glycine (final concentration 125 mM) and incubating with gentle agitation for 5 min at 25°C.

5   Centrifuge at 974 *g* for 2 min at 4°C and remove supernatant.

6   Resuspend and wash cross-linked infected red blood cells with 10 ml DPBS at 4°C.

7   Centrifuge at 974 *g* for 2 min at 4°C and remove supernatant.

8   Lyse infected red blood cells in 10 ml 0.15% saponin (Sigma S7900) in DPBS at 4°C for 5–10 min (until liquid becomes transparent).

9   Centrifuge at 3,220 *g* for 5 min at 4°C and remove supernatant.

10  Wash the parasites with 12 ml DPBS at 4°C.

11  Centrifuge at 3,220 *g* for 5 min at 4°C and remove supernatant.

12  Repeat steps 10–11.

13  If the supernatant is clear and not red, remove it and proceed to the next step or snap freeze and store at −80°C. If it is not clear, wash the pellet again with DPBS.

14  Resuspended parasites in 2 ml of lysis buffer (10 mM HEPES pH 8, 10 mM KCl, 0.1 mM EDTA pH 8, PI) at 4°C and incubate with gentle agitation at 4°C for 30 min.

15   Add IGEPAL CA-630 to a final concentration of 0.25% and lyse cells with a prechilled dounce homogenizer (usually 200 strokes are sufficient for ring stage parasites).

16   Transfer extracts to microcentrifuge tubes and centrifuge for 10 min at 13,500 $g$ at 4°C.

17   Remove supernatant and resuspend the pellet in 300 µl SDS lysis buffer (50 mM Tris–HCl pH 8, 10 mM EDTA pH 8, 1% SDS, PI) at 4°C.

18   Transfer liquid to a 1.5 ml sonication tube (Diagenode C30010016) and sonicate for 12 min total (24 cycles of 30 s on/off) in a Diagenode Pico Bioruptor at 4°C. After sonication, run a small amount of sample on a DNA gel or an Agilent 2100 Bioanalyzer to make sure the average DNA fragment size is 250 bp.

19   Centrifuge sonicated extracts for 10 min at 13,500 $g$ at 4°C.

20   Using the DynaMag, wash antibody-conjugated Dynabeads twice with 1 ml ChIP dilution buffer and resuspend in 100 µl of ChIP dilution buffer at 4°C.

21   Dilute 120 µl of supernatant 1:10 in ChIP dilution buffer at 4°C.

22   Add 30 µl of supernatant to 170 µl of elution buffer (50 mM Tris–HCl pH 8, 10 mM EDTA pH 8, 1% SDS) and keep as "Input" at −20°C.

23   Add washed, antibody-conjugated Dynabeads to the diluted chromatin sample and incubate overnight with rotation at 4°C.

24   Collect beads on a DynaMag, remove supernatant, and wash for 5 min with gentle rotation with 1 ml of the following buffers, sequentially:

- Low salt wash buffer (20 mM Tris–HCl pH 8, 150 mM NaCl, 2 mM EDTA pH 8, 1% Triton X-100, 0.1% SDS) at 4°C.
- High salt wash buffer (20 mM Tris–HCl pH 8, 500 mM NaCl, 2 mM EDTA pH 8, 1% Triton X-100, 0.1% SDS) at 4°C.
- LiCl wash buffer (10 mM Tris–HCl pH 8, 250 mM LiCl, 1 mM EDTA pH 8, 0.5% IGEPAL CA-630, 0.5% sodium deoxycholate) at 4°C.
- TE wash buffer (10 mM Tris–HCl pH 8, 1 mM EDTA pH 8) at room temperature.

25   Collect beads on a DynaMag, remove supernatant, and resuspend the beads in 205 µl of elution buffer.

26   Incubate the beads for 30 min at 65°C with agitation.

27   Collect beads on a DynaMag and transfer 200 µl of eluate to a different tube. This is the "ChIP" sample.

28   De-cross-link input and ChIP samples for ∼ 10 h at 65°C.

29   Add 200 µl of TE buffer to each sample.

30   Add 8 µl RNaseA (Thermo Fisher EN0531) to each sample (final concentration of 0.2 mg/ml) and incubate for 2 h at 37°C.

31   Add 4 µl Proteinase K (New England Biolabs P8107S) to each sample (final concentration of 0.2 mg/ml) and incubate for 2 h at 55°C.

32   Add 400 µl phenol:chloroform:isoamyl alcohol (25:24:1) to each sample, vortex, and separate phases by centrifugation for 10 min at 13,500 $g$ at 4°C.

33   Keep the top (aqueous) layer, add 30 µg glycogen (Thermo Fisher 10814) and NaCl (200 mM final concentration), and mix.

34   Add 800 µl 100% EtOH at 4°C to each sample and incubate at −20°C for 30 min.

35   Pellet DNA by centrifugation for 10 min at 13,500 $g$ at 4°C.

36   Wash pellet with 500 µl 80% EtOH at 4°C and centrifuge for 5 min at 13,500 $g$ at 4°C.

37   Remove the EtOH and dry pellet at 25°C.

38   Resuspend dried pellet in 30 µl 10 mM Tris–HCl, pH 8.

### dCas9 chromatin immunoprecipitation sequencing and analysis

Libraries for sequencing were made with the MicroPlex library preparation kit (Diagenode C05010014). Libraries were sequenced on the NextSeq 500 platform (Illumina). Sequenced reads (150 bp paired end) were mapped to the *P. falciparum* genome (Gardner *et al*, 2002) (plasmoDB.org, version 3, release 29) using "bwa mem" (Li & Durbin, 2009) allowing a read to align only once to the reference genome (option "–c 1"). Alignments were subsequently filtered for duplicates and a mapping quality ≥ 20 using samtools (Li *et al*, 2009). For ChIP-seq coverage plots, deeptool's bamCompare (Ramírez *et al*, 2016) was used to normalize the read coverage per genome position (option "–bs 1") in the respective input and ChIP samples to the total number of mapped reads in each library (option "–normalizeUsingRPKM"). Normalized input coverage per bin was subtracted from the ChIP values (option "–ratio subtract"), and coverage plots were visualized in the Integrative Genomics Viewer (Robinson *et al*, 2011). For genome-wide coverage plots, the same approach was used as above but with a bin size of 1,000 nt. Background levels of unspecific dCas9 binding were further removed by subtracting normalized coverage of the non-targeted dCas9 control sample from the normalized promoter or intron coverages and visualized using circos (Krzywinski *et al*, 2009).

Significant peaks of dCas9 binding sites were identified using macs2 (version 2.2.6) (Zhang *et al*, 2008) "callpeak" using default settings and options "–nomodel", "–extsize150" and "–B" set. A Benjamini–Hochberg-adjusted *P*-value (i.e., *q*-value) was calculated for each peak. To compare the ChIP enrichment of dCas9 in the *var* gene intron- or promoter-targeted strains to the non-targeted control strain, a likelihood ratio of enrichment comparing promoter/control (Table EV1) and intron/control (Table EV2) was calculated using the output pileup files from "callpeak" and the macs2 command "bdgdiff" with default settings. One replicate was performed per strain.

### dCas9 RNA immunoprecipitation sequencing and analysis

dCas9 RIP-seq was performed with the same protocol for dCas9 ChIP-seq (above) with the following modifications. RNasin RNase inhibitor (Promega N2511) was added to all lysis and wash buffers. A control immunoprecipitation was performed using 1 µg of rabbit IgG (Diagenode C15410206) conjugated to 25 µl of Protein G Dynabeads (Invitrogen 10004D). After bead incubation, samples were washed and eluted as above and then incubated for 30 min at 70°C to reverse cross-links. One microliter of DNase Turbo (Thermo Fisher AM2238) was added, and samples were incubated for 20 min at 37°C. Six hundred microliters of Trizol LS (Thermo Fisher 10296010) were mixed with the sample, followed by 160 µl chloroform. Samples were vortexed and phases were separated by centrifugation for 10 min at 13,500 $g$ at 4°C. The aqueous (top) layer was then purified with the RNeasy MinElute Cleanup Kit (Qiagen 74204). RNA libraries were prepared with the TruSeq stranded mRNA LT library prep kit (Illumina 20020594) without mRNA selection, ribosomal RNA depletion, or fragmentation. Libraries were sequenced, and data were treated as above for ChIP. One replicate

was performed for anti-HA and IgG immunoprecipitations in non-clonal bulk cultures at 12 hpi of non-targeted dCas9 and *var* promoter-targeted dCas9 strains.

For dCas9 RIP-seq analysis of the *var* promoter-targeted dCas9, read coverage in the anti-HA and IgG samples was calculated using deepTools "bamCompare" (Ramírez *et al*, 2016) in 10 nt windows (option "–bs 10") and normalized to the total number of mapped reads in each library (option "–normalizeUsingRPKM"). The normalized coverage in each bin of the IgG sample was then subtracted from the anti-HA sample (option "–ratio subtract"). Promoter/control fold enrichment of each IgG-normalized 10 nt window was calculated using macs2 "bdgcmp" (Zhang *et al*, 2008). For genome-wide coverage plots, the fold enrichment as calculated by macs2 "bdgcmp" was averaged over the gene body and visualized using circos (Krzywinski *et al*, 2009).

### Stranded RNA sequencing and analysis

Infected RBCs containing synchronized (14 hpi $\pm$ 3 h) parasites were lysed in 0.075% saponin (Sigma S7900) in DPBS at 37°C. The parasite cell pellet was washed once with DPBS and then resuspended in 700 µl QIAzol reagent (Qiagen 79306). Total RNA was extracted using the miRNeasy Mini Kit (Qiagen 217004), including an on-column DNase I digestion according to the manufacturer's protocol. RNA samples were depleted of rRNA with the Dynabeads mRNA DIRECT Kit (Thermo Fisher 61012), and libraries were prepared with the TruSeq Stranded mRNA LT Sample Prep Kit (Illumina 20020594). Libraries were sequenced on the NextSeq 500 platform (Illumina). Sequenced reads (150 bp paired end) were mapped, aligned, and visualized as described above for ChIP-seq.

### Immunoprecipitation of dCas9 for mass spectrometry analysis

1   Wash and conjugate beads as described above for ChIP-seq. Ten milligrams of Protein G Dynabeads (Invitrogen 10004D) conjugated to 10 µg of anti-HA antibody (Abcam ab9110) are needed per immunoprecipitation.

2   Cross-link and perform saponin lysis with synchronized parasites (14 hpi $\pm$ 3 h) as described above for ChIP-seq. $3 \times 10^9$ cross-linked parasites were collected for each immunoprecipitation, with four replicates per strain (non-targeted, promoter-targeted, or intron-targeted dCas9).

3   Resuspend pellet ($3 \times 10^9$ parasites) in 10 ml of lysis buffer (10 mM Tris–HCl pH 7.5, 1 mM EDTA, 0.5% IGEPAL CA-630, PI) at 4°C and incubate with rotation for 30 min at 4°C.

4   Centrifuge extracts for 8 min at 380 *g* at 4°C and remove the supernatant.

5   Resuspend the pellet in 10 ml nuclear lysis buffer (10 mM Tris–HCl pH 7.5, 500 mM NaCl, 1 mM EDTA, 1% IGEPAL CA-630, 0.5% sodium deoxycholate, PI) at 4°C and incubate for 10 min at 4°C, vortexing every ~3 min.

6   Centrifuge lysates for 8 min at 380 *g* at 4°C and remove supernatant.

7   Resuspend the pellet in 1.8 ml chromatin shearing buffer (10 mM Tris–HCl pH 7.5, 150 mM NaCl, 1 mM EDTA, 0.1% sodium deoxycholate, 0.1% SDS, PI) at 4°C and transfer to 1.5 ml sonication tubes (300 µl per tube, Diagenode C30010016).

8   Sonicate for 12 min total (24 cycles of 30 s on/off) in a Diagenode Pico Bioruptor at 4°C.

9   Centrifuge lysate for 10 min at 13,500 *g* at 4°C and keep supernatant.

10  Dilute all supernatant of the chromatin lysate 1:10 in wash buffer (20 mM Tris–HCl pH 7.5, 150 mM NaCl, 1 mM EDTA, 0.1% IGEPAL CA-630, 10% glycerol, PI) at 4°C.

11  Add antibody-conjugated beads to the diluted chromatin sample and incubate overnight with rotation at 4°C.

12  For each immunoprecipitation, use three microcentrifuge tubes to collect the beads from the total volume, 1 ml at a time using the DynaMag. For example, add 1 ml to the tube on the magnet, collect the beads, remove supernatant, and repeat.

13  Wash the beads in each tube 3 × 1 ml wash buffer for 5 min with rotation at 4°C.

14  Wash the beads in each tube with 1 ml TE buffer for 5 min with rotation at 4°C.

15  Collect beads on the DynaMag and wash three times with 100 µl 25 mM ammonium bicarbonate ($NH_4HCO_3$) without removing the beads from the magnet.

16  Remove tube from the magnet and resuspend beads in 100 µl 25 mM $NH_4HCO_3$.

17  Perform on-bead digestion for 1 h with 0.6 µg of trypsin/LysC (Promega V5071).

18  Load samples onto homemade C18 StageTips for desalting and elute peptides using 40/60 MeCN/$H_2O$ + 0.1% formic acid.

19  Vacuum concentrate samples to dryness.

20  Perform online chromatography with an RSLCnano system (Thermo Fisher, Ultimate 3000) coupled online to a Q Exactive HF-X with a Nanospray Flex ion source (Thermo Fisher). Peptides were first trapped on a C18 column (75 µm inner diameter × 2 cm; nanoViper Acclaim PepMap™ 100, Thermo Fisher 164942) with buffer A (2/98 MeCN/$H_2O$ in 0.1% formic acid) at a flow rate of 2.5 µl/min over 4 min. Separation was then performed on a 50 cm × 75 µm C18 column (nanoViper Acclaim PepMap™ RSLC, 2 µm, 100 Å, Thermo Scientific 164535) regulated to a temperature of 50°C with a linear gradient of 2–30% buffer B (100% MeCN in 0.1% formic acid) at a flow rate of 300 nl/min over 91 min.

21  Perform MS full scans in an ultrahigh-field Orbitrap mass analyzer in ranges $m/z$ 375–1,500 with a resolution of 120,000 at $m/z$ 200. The top 20 intense ions were subjected to Orbitrap for further fragmentation via high energy collision dissociation (HCD) activation and a resolution of 15,000 with the intensity threshold kept at $1.3 \times 10^5$. Ions with charge state from 2+ to 6+ were selected for screening. Normalized collision energy (NCE) was set at 27 with a dynamic exclusion of 40 s.

22  Perform analysis.

•   For identification, the data were searched against the *P. falciparum* FASTA database (PlasmoDB-36 Pfaciparum3D7 AnnotatedProtein containing Cas9 and the common contaminants) using Sequest HF through proteome discoverer (Thermo Fisher version 2.2). Enzyme specificity was set to trypsin, and a maximum of two missed cleavage sites were allowed. Oxidized methionine and N-terminal acetylation were set as variable modifications. Maximum allowed mass deviation was set to 10 ppm for monoisotopic precursor ions and 0.02 Da for MS/MS peaks.

•   The resulting files were further processed using myProMS (ver-

sion 3.9) (Poullet *et al*, 2007). For identification, FDR was computed using Percolator (Spivak *et al*, 2009) and was set to 1% at the peptide-level for the whole study. The label-free quantification was performed by peptide Extracted Ion Chromatograms (XICs) computed with MassChroQ version 2.2 (Valot *et al*, 2011). For protein quantification, XICs from proteotypic peptides shared between compared conditions (TopN matching) with two missed cleavages were used. Median and scale normalization was applied on the total signal to correct the XICs for each replicate ($n = 4$). Two replicates of the promoter-targeted dCas9 sample were excluded from the analysis due to significantly lower amounts of dCas9 and total protein detected, which would lead to a biased analysis (Fig EV3A and Table EV4).

- To estimate the significance of the change in protein abundance, a linear model (adjusted on peptides and replicates) based on two-tailed *t*-test was performed and *P*-values were adjusted using Benjamini–Hochberg FDR procedure. Proteins with at least 1.5-fold enrichment, *P*-value ≤ 0.05, and at least three total peptides in all replicates were considered significantly enriched in sample comparisons.

### ISWI immunoprecipitation and mass spectrometry

An ISWI-3HA-ribo clone ($n = 3$ technical replicates) and wild-type culture (as a negative control) were synchronized. At 36 hpi, each culture ($1.5 \times 10^9$ parasites) was centrifuged and RBCs were lysed with six volumes of 0.15% saponin in DPBS for 5 min at 4°C. Parasites were centrifuged at 4,000 *g* for 5 min at 4°C, and the pellet was washed twice with DPBS at 4°C. Parasites were then cross-linked with 1 ml 0.5 mM dithiobissuccinimidyl propionate (DSP; Thermo Fisher 22585) in DPBS for 60 min at 37°C (as in (Mesén-Ramírez *et al*, 2016)). Cross-linked parasites were washed once with DPBS and then lysed with 10 volumes of RIPA buffer (10 mM Tris–HCl pH 7.5, 150 mM NaCl, 0.1% SDS, 1% Triton) containing protease and phosphatase inhibitor cocktail (Thermo Fisher 78440) and 1 U/µl of Benzonase (Merck 71206). The lysates were cleared by centrifugation at 16,000 *g* for 10 min at 4°C. Supernatants were incubated with 25 µl of anti-HA Dynabeads (Thermo Fisher 88836) overnight with rotation at 4°C. Beads were collected with a magnet and washed five times with 1 ml RIPA buffer, then five times with 1 ml DPBS, and then once with 1 ml 25 mM $NH_4HCO_3$ (Sigma 09830). The beads were reduced with 100 mM dithiothreitol (Sigma D9779), alkylated with 55 mM iodoacetamide (Sigma I1149), and subjected to on-bead digestion using 1 µg of trypsin (Thermo Fisher 90059). The resulting peptides were desalted using C18 ziptips (Merck ZTC04S096) and sent for MS analysis.

Peptides were separated by reverse phase HPLC (Thermo Fisher Easy-nLC1000) using an EASY-Spray column, 50 cm × 75 µm ID, PepMap RSLC C18, 2 µm (Thermo Fisher ES803A) over a 90-min gradient before nanoelectrospray using a Q Exactive HF-X mass spectrometer (Thermo Fisher). The mass spectrometer was operated in a data-dependent mode. The parameters for the full scan MS were as follows: resolution of 60,000 across 350–1,500 *m/z*, AGC $1e^3$ (as in (Kensche *et al*, 2016)), and maximum injection time (IT) 50 ms. The full MS scan was followed by MS/MS for the top 15 precursor ions in each cycle with an NCE of 28 and dynamic exclusion of 30 s and maximum IT of 300 ms. Raw mass spectral data files (.raw) were searched

using Proteome Discoverer (Thermo Fisher) and Mascot (version 2.4.1) (Perkins *et al*, 1999). Mascot search parameters were as follows: 10 ppm mass tolerance for precursor ions; 0.8 Da for fragment ion mass tolerance; two missed cleavages of trypsin; fixed modification was carbamidomethylation of cysteine; and variable modifications were methionine oxidation, CAMthiopropanoyl on lysine or protein N-terminal, and serine, threonine, and tyrosine phosphorylation. Only peptides with a Mascot score greater than or equal to 25 and an isolation interference less than or equal to 30 were included in the data analysis.

### Phylogenetic analysis of SNF2 domain-containing chromatin remodelers

SNF2 domains were extracted from the full protein sequences following the "SNF2 family N-terminal domain" Pfam annotation (Pfam ID: PF00176) using bedtools getfasta (Quinlan & Hall, 2010). Multiple sequence alignments were performed using MAFFT with default settings. Gaps were removed with trimAl (Capella-Gutiérrez *et al*, 2009), and the best phylogenetic model (i.e., Le Gascuel [LG]) was calculated using ProtTest3 (Darriba *et al*, 2011). A maximum likelihood phylogenetic tree was constructed using MEGA (v7) (Kumar *et al*, 2016) with the LG model and 1,000 bootstrap replicates. The bootstrap consensus trees were visualized in FigTree (v1.4.3, http://tree.bio.ed.ac.uk/software/figtree/).

### Parasite growth assay

Parasite growth was measured as described previously (Vembar *et al*, 2015). Briefly, a clone of ISWI-3HA-ribo was tightly synchronized and diluted to 0.2% parasitemia (5% hematocrit) at ring stage using the blood of two different donors separately. Each culture was split, and glucosamine (Sigma G1514) was added (2.5 mM final concentration) to one half. The growth curve was performed in a 96-well plate (200 µl culture per well) with three technical replicates per condition per blood. Parasitemia was measured every 24 h by staining parasite nuclei with SYBR Green I (Sigma S9430) and quantifying infected red blood cells with a Guava easyCyte flow cytometer (Merck Millipore).

### Estimation of cell cycle progression

RNA-seq-based cell cycle progression was estimated in R by comparing the normalized expression values (i.e., RPKM, reads per kilobase per exon per one million mapped reads) of each sample to the microarray data from Bozdech *et al* (2003) (Data ref: Bozdech *et al*, 2003) using the statistical model as in Lemieux *et al* (2009).

### Differential gene expression analysis

A clone of ISWI-3HA-ribo was synchronized and split into two cultures. Glucosamine (2.5 mM final concentration, Sigma G1514) was added to one culture at 36 hpi, and parasites were harvested 24 h later during ring stage (12 hpi). RNA sequencing reads for three technical replicates of the glucosamine-treated and two technical replicates of the untreated ISWI-3HA-ribo clone were mapped to the *P. falciparum* genome and quality filtered as described above for ChIP-seq and RNA-seq. Strand-specific gene counts were calculated using htseq-count (Anders *et al*, 2015). Differential gene expression analysis was performed using DESeq2 (Love *et al*, 2014) with significantly differentially expressed genes featuring a Benjamini–Hochberg-adjusted *P*-value (i.e., *q*) ≤ 0.05. MA plots were generated

using the "baseMean" (mean normalized read count over all replicates and conditions) and "$log_2$FoldChange" values (glucosamine-treated over control) as determined by DESeq2. RPKM values were calculated in R using the command rpkm() from the package edgeR (Robinson *et al*, 2010). Gene Ontology enrichments were calculated using the built-in tool at PlasmoDB.org (Aurrecoechea *et al*, 2017).

### ISWI and H3K9ac ChIP sequencing and analysis

Synchronous parasites (12 hpi ± 3) were treated as above for the dCas9 ChIP with minor differences. $5 \times 10^8$ parasites were cross-linked and lysed in 4 ml ChIP lysis buffer. Chromatin was sonicated in 3.6 ml total ChIP SDS lysis buffer. For the ISWI-HA ChIP, 8 μg DNA (chromatin) was diluted and incubated with 8 μg anti-HA antibody (Abcam ab9110) conjugated to 200 μl Protein G Dynabeads (Invitrogen 10004D). For the H3K9ac ChIP, 2 μg DNA (chromatin) was diluted and incubated with 2 μg anti-H3K9ac antibody (Millipore 07-352) conjugated to 50 μl Protein G Dynabeads (Invitrogen 10004D).

Sequencing libraries were prepared, sequenced, and processed as with the dCas9 ChIP samples. Fold enrichment over input was calculated using deeptool's bamCompare (Ramírez *et al*, 2016) (option "–ratio ratio") in windows of 1 nt (option "–bs 1"), and coverage plots were visualized in the Integrative Genomics Viewer (Robinson *et al*, 2011). Meta-gene plots were visualized using deeptool's plotProfile over a region 1.5 kb upstream and downstream of the translation start and stop site, respectively, using the quality-filtered bam files as input.

## Data availability

The data sets generated in this study are available in the following databases:

- ChIP-seq data: NCBI BioProject accession #PRJNA529754 (http://ncbi.nlm.nih.gov/bioproject/PRJNA529754)
- RNA-seq data: NCBI BioProject accession #PRJNA529754 (http://ncbi.nlm.nih.gov/bioproject/PRJNA529754)
- dCas9 Proteomics data: PRIDE PXD013131 (http://www.ebi.ac.uk/pride/archive/projects/PXD013131)
- ISWI-3HA Proteomics data: https://chorusproject.org/pages/dashboard.html#/projects/all/1645/experiments

Expanded View for this article is available online.

## Acknowledgements

This work was supported by the European Research Council (grant PlasmoSilencing 670301 to A.Sc.); the Agence Nationale de la Recherche (grant ANR-11-LABEX-0024-01 ParaFrap to A.Sc.); the "Région Ile-de-France" (grants 2013-2-EML-02-ICR-1 and 2014-2-INV-04-ICR-1 to D.L.); the Fondation pour la Recherche Médicale (grant DGE20121125630 to D.L.); the Academic Research Fund (Tier 2) of the Ministry of Education, Singapore (MOE2018-T2-2-131 to P.R.P.); and the Merlion Project (6.11.18 to P.R.P). Work in the laboratories of P.R.P. and P.C.D. was funded by the National Research Foundation of Singapore through the Singapore-MIT Alliance for Research and Technology Antimicrobial Resistance Interdisciplinary Research Group. Proteomics work was performed in part in the Center for Environmental Health Sciences BioCore, which is supported by Center grant P30-ES002109 from the National Institute of Environmental Health Sciences. J.M.B. was supported by a European Molecular Biology Organization long-term postdoctoral fellowship (EMBO ALTF 180-2015), the Institut Pasteur Roux-Cantarini postdoctoral fellowship, and a ParaFrap fellowship. S.B. was supported by a European Molecular Biology Organization long-term postdoctoral fellowship (EMBO ALTF 1444-2016) and advanced fellowship (EMBO aALTF 632-2018). A.Si. was supported by the Singapore-MIT Alliance (SMA) Graduate Fellowship from the Ministry of Education of Singapore. The authors would like to acknowledge the use of mass spectrometer facilities at A*STAR Institute of Molecular and Cell Biology in the laboratory of Dr. Radoslaw Sabota with the aid of

Dr. Wint Wint Phoo. The authors would like to thank Valentin Sabatet at the Institut Curie for his help with the interpretation of the dCas9 proteomics data.

## Author contributions

JMB, SB, and ASc conceptualized the project and conceived experiments. AC cloned the optimized pL7 plasmid. FD carried out the mass spectrometry experimental work for dCas9 enChIP, and DL supervised mass spectrometry and data analysis. SB performed DNA/RNA sequencing bioinformatic analysis. ASi performed the ISWI immunoprecipitation mass spectrometry and analysis. PRP, PCD, and ASc supervised and helped interpret analyses. JMB performed all other experiments and wrote the manuscript. All authors discussed and approved the manuscript.

## Conflict of interest

The funders had no role in the study design, data collection and interpretation, or the decision to submit the work for publication. The authors declare that they have no conflict of interest.

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
