## [Review Process File · Molecular Systems Biology]

Exploring the virulence gene interactome with CRISPR/dCas9 in the human malaria parasite

Jessica Bryant, Sebastian Baumgarten, Florent Dingli, Damarys Loew, AMEYA SINHA, Aurelie Claes, Peter Preiser, Peter Dedon, and Artur Scherf

DOI: [10.15252/msb.20209569](https://doi.org/10.15252/msb.20209569)

Corresponding author(s): Jessica Bryant (jessica.bryant@pasteur.fr)

Review Timeline:

Submission Date:	9th Mar 20
Editorial Decision:	24th Apr 20
Appeal Received:	29th Apr 20
Editorial Decision:	6th May 20
Revision Received:	14th Jun 20
Editorial Decision:	5th Jul 20
Revision Received:	17th Jul 20
Accepted:	22nd Jul 20

Editor: Jingyi Hou

Transaction Report:

We have now received all the reports and as you will see below, the reviewers raise substantial concerns on your work, which unfortunately preclude its publication in *Molecular Systems Biology*.

The reviewers acknowledge the potential interest of the study and appreciate the novelty of the presented CRISPR-Cas9-based approach. However, while reviewer #3 is relatively more supportive, both reviewer #1 and #2 raise significant concerns with regards to the lack of key controls, validations and limited provision of novel biological insights. In particular, during our pre-decision cross-commenting process (in which the reviewers are given the chance to make additional comments, including on each other's reports), reviewer #1 and #2 indicate that they think addressing these concerns are essential to make the study conclusive. Specifically, reviewer #2 added "Generating the additional controls will require a large amount of new experimental work (at least 6 months should be allowed), but I believe these controls are essential even if the authors decided to submit the manuscript to a lower profile journal. Same as Reviewer 1, I am usually reluctant to request a large set of additional experiments, but in this case I believe that the additional controls are essential for a correct interpretation of the data. Without these experiments, the conclusions are not well supported."

Under these circumstances and given that the concerns raised by the reviewers are substantial and are unlikely to be addressable within the scope of a major revision (which is usually within a three-month period), we see no other choice than to return the manuscript with the message that we cannot offer to publish it.

Nevertheless, as the reviewers did acknowledge that the topic of the study is potentially relevant, we would not be opposed to considering a substantially revised and extended manuscript based on this work, provided that the issues raised by the reviewers can be convincingly addressed. Some of the more essential issues that would need to be addressed include:

- Key control experiments should be included (especially in the proteomics experiments) to make the study more conclusive. Additional experiments and analyses are required to better support the data, as suggested by the reviewers.
- Reviewer #1 and #2 also pointed out that one of the major issues at this point is the insufficient follow-up experiments regarding the role of ISWI. These concerns need to be addressed to improve the conclusiveness and to enhance the level of biological insights provided by the study.

All three reviewers provide constructive suggestions on how to address the points above and improve the study. We understand that this requires a significant investment and may prove challenging. I would be happy to look at a preliminary point-by-point response delineating how the

issues raised can be addressed, so that we can work together on how to move forward. We also recognize that thoroughly addressing the referees' concerns would involve substantial further experiments with unclear outcome and we understand if in light of the substantial revisions required, you prefer to submit your study elsewhere.

A resubmitted work would have a new number and receipt date. It will be editorially evaluated afresh and its novelty will be re-assessed at the time of submission. As you probably understand, we can give no guarantee about its eventual acceptability. If you do decide to follow this course then we would ask you to enclose with your re-submission a point-by-point response to the points raised in the present review.

I am sorry that the review of your work did not result in a more favorable outcome on this occasion, but I hope that you will not be discouraged from sending your work to Molecular Systems Biology in the future. In any case, thank you for the opportunity to examine this work.

REFEREE REPORTS

Reviewer #1:

This manuscript uses an innovative approach, dCas9 targeting and immunoprecipitation, to attempt to identify proteins associated with the regulatory elements of var genes, the multi-gene family that are responsible for Plasmodium falciparum cytoadherence and therefore play a key role in malaria pathogenesis. Control of var gene expression is a significant area of focus for malaria biology and this novel piece of work identifies new potential components of what is undoubtedly a complex molecular system. The manuscript is well written and well referenced, with methods described in commendable detail, and data is being made broadly open access, also to be commended. Enthusiasm for the manuscript in its current form is however diminished by issues with control, quantification and validation, which make the key conclusions currently less than convincing. These need addressing or clarifying in a revised version.

Major points

1. dCas9 targeting control. Distinguishing between var-specific factors and general chromatin associated factors was surely one of the key goals of the work - that is certainly how the research goal is positioned in the Introduction. If that is the case, having a control line that helps to distinguish var-specific signal from general noise would seem to be key. The control line is however described as "a non-specific sgRNA that does not target any genomic sequence". It's not quite

clear what this means, and I could not find any reference to the specific sgRNA sequence in the Methods (apologies if I overlooked it). Was the control sgRNA deliberately designed to not bind to any sequence in the *P. falciparum* genome? If so, how was this achieved bioinformatically? If not, how was it designed? More importantly, what was the rationale for this? Surely to identify var-specific factors, the key control would be to target dCas9 to a different chromosomal location, not associated with var genes, but still actively transcribed during the 12-14 hour life cycle stage? Repeating the IP and mass spec with such a control would of course be a very large amount of work, and the lack of it doesn't completely invalidate the findings, but at the very least, the rationale for why this approach was taken would be helpful, and a much clearer explanation of how and why the control sgRNA was chosen.

2. Western blot validation. On a similar issue, the Western blot shown in Fig 1C establishes that one of the targeted dCas9 constructs localises primarily to chromatin, but it is not specified which one (var-promoter or intron) and critically the control targeted dCas9 is not shown to establish whether it is even nuclear in localisation. Western blots with fractionated material is needed from all three lines to make it clear what is actually being compared in the subsequent ChIP-seq and mass spec experiments. If the control line does not have nuclear targeted dCas9, it is not an effective control for subsequent experiments.

3. Mass spectrometry interpretation. It is not clear why mass spectrometry was not carried out for the control line. Given the number of cytoskeleton-associated proteins enriched with the intron targeted dCas9 in particular, there is obviously some concern about non-specific pulldown of abundant proteins. This is an issue with any mass spec study of course, which is why controls are so important. The increase of such abundant, potentially non-specific, proteins in the intron targeted line may be linked to the fact that the ChIP-seq data showed that binding of dCas9 was less enriched at the intron targeted sequences than the promoter targeted sequence, implying a higher level of background binding. This emphasises again that dCas9 targeted to a different gene would seem to have been the perfect control for the mass spectrometry study. While reluctant to ask for another large set of experiments, it is not clear without such data how the mass spectrometry findings can be interpreted clearly. Presumably also the approach of looking for proteins enriched at one or other location also overlooks the potential to identify proteins that are found at both, yet are still var-specific.

4. Validation. ISWI is chosen as the sole target for validation, and immunoprecipitation of ISWI-3HA co-purifies several proteins that were also identified in the initial mass spec. This is encouraging, but it is also critical to establish whether ISWI-3HA localises to var promoters, either by CHIP-seq and/or an immunolocalization approach. This will distinguish between ISWI being a background interaction, or one that is specifically associated with var gene regulation, which is after all the goal of the project.

Minor points

1. Synchronisation. Tight synchronisation is obviously key to the comparability between replicates for the mass spec studies. While the synchronisation method is described, was synchrony actually quantitatively or even semi-quantitatively measured in each replicate, for example by qRT-PCR, or even simple morphology, to confirm that the replicates were broadly comparable?

2. ISWI-3HA IP control. Why were wildtype parasites used as an IP control, which would presumably would not yield any immunoprecipitated protein at all, as there is no anti-HA target in the line? The lack of any immunoprecipitate will significantly reduce background, and also therefore significantly reduce the ability to distinguish signal from noise in the ISWI-3HA IP material. Another HA-tagged protein would be a much better control - given that they also have generated dCas9-3HA, surely this would have provided a better control for this IP, even without gRNA targeting?

Reviewer #2:

In this manuscript, Bryant et al. develop a method based on CRISPR-dCas9 targeting and quantitative proteomics to identify proteins associated in vivo with specific regions of the *Plasmodium falciparum* genome. This is a major technical development that will enable the identification of the malarial regulatory proteins associated with specific loci. The authors carefully demonstrate the specificity of the binding of dCas9 to the intended loci using ChIP-seq analysis. They use the method to characterize the proteins that bind the promoter and intron of var virulence genes, which results in the identification of a large number of differentially bound proteins. One of these proteins, the chromatin remodeler ISWI, is further characterized using knockdown and co-immunoprecipitation approaches.

The establishment of the CRISPR-based immunoprecipitation-proteomics method is a big advance for the field, but the manuscript provides limited biological insight on the regulation of var genes. Furthermore, there are important issues with the interpretation of some of the experiments and a key control is missing.

Major comments.

1. An important limitation of this study is that the transcriptional state of the targeted var genes has not been unambiguously determined, and consequently it is unclear whether the proteins identified associate with the chromatin of the active or the silenced state. If dCas9 binding does not alter the normal regulation of var genes, mutually exclusive expression implies that out of the 15-19 genes targeted, only one (or none in parasites in which a non-targeted var gene is expressed) is in an active state in each parasite. Thus, the majority of factors identified are likely associated with the silenced state. However, the authors claim that in the promoter-targeted dCas9 line, the dCas9-bound var genes are expressed at higher level, based on RIP-seq and RNA-seq data. Unfortunately, the data is not shown in an informative way. Only two specific genes are shown in fig 2c (without showing the non-targeted dCas9 control strain for comparison), and in the supplementary tables, RNA-seq data is only shown for a subset of the genes with differential RIP-seq signal. Whether or not other var genes (those not bound by dCas9) are also expressed at higher levels in the promoter-targeted dCas9 strain is not described. The RNA-seq and RIP-seq data should be presented in a way that can be evaluated by the reader. The authors should provide, at least, complete supplementary datasets, and a bar chart including all targeted and non-targeted var genes (analogous to the chart in fig 4d).

The suggestion by the authors that dCas9 binds to active promoters is not well-supported: even if dCas9 binds preferentially to active promoters, this wouldn't result in increased expression of the target genes as determined by RNA-seq if dCas9 binding does not alter expression. If, instead, the authors want to suggest that dCas9 binding results in increased expression of the target genes and disrupts mutually exclusive expression, such that the majority of dCas9-bound var genes are simultaneously active, this would need to be supported by clearly presented data, and it may represent an important concern for the interpretation of the data obtained with this system.

There are specific systems that enable the study of var promoters in a controlled active or silenced state. The authors should investigate the effect of dCas9 binding on var gene expression in clones expressing a single specific var gene, or in transgenic lines in which a var promoter controls the expression of a drug resistance marker. Identification of proteins bound to var control elements at stages when the genes are not expressed (e.g. schizont stage) may also identify proteins specifically associated with the silenced state.

2. The role of ISWI as a specific activator of var genes is not well demonstrated. First, the magnitude of the reduction of expression in the knockdown line is modest and may be explained by delayed life cycle progression as a consequence of glucosamine addition, or as a consequence of the reduced expression of ISWI. These possibilities should be excluded with appropriate control experiments. Second, over 100 genes show larger changes in expression than the var gene upon ISWI knockdown. Therefore, unless additional evidence is provided, ISWI appears to be a likely constitutive component of the basal transcriptional machinery, rather than a specific regulator of var gene expression. The argument in the Discussion (page 14) that if ISWI was required for general processes a more dramatic phenotype would be expected is not valid, because this may depend on the level of downregulation (which, according to fig 4a, is modest).

3. My other major concern is about the quantitative proteomics-based identification of factors bound to the promoter or the intron. Unfortunately, control experiments with the non-targeted dCas9 strain were not performed. This control would be important to distinguish which of the many proteins identified are specifically bound to var genes, and which were non-specifically immunoprecipitated. Without this control, it is impossible to determine which of the >1,000 proteins actually participate in var regulation. The current analysis is focused on differential binding between promoter and intron, but this cannot replace an appropriate negative control with a non-targeted dCas9 strain. Furthermore, factors that bind to both the promoter and the intron may also be of interest.

The analysis is selective, as it focuses on proteins involved in chromatin remodeling without even demonstrating that this functional category is enriched among the proteins identified, and many other highly enriched proteins are not discussed. An unbiased analysis of the data, in addition to the focus on specific candidate regulators, should be presented.

Minor comments.

Having generated the transgenic line expressing HA-glms-tagged ISWI, the genome-wide distribution of tagged ISWI could be analyzed by ChIP-seq. This would help to clarify whether ISWI plays a preferential role in var regulation or is a general factor necessary for transcription of most genes.

Please provide a further analysis of the genes differentially expressed genes in the ISWI knockdown line. Are they among the most highly expressed genes at the ring stage? Are they genes with peak expression at other stages?

Page 6. TSS is commonly used to refer to Transcription Start Sites. Using it to refer to the translation start site (ATG start codon) can be misleading. The scheme in Fig. 1A should reflect that promoter-targeted dCas9 is closer to the start codon than to the transcription start site (40 bp from the former and up to 2 Kb from the latter).

Page 7. The off-target genes and 'unpredicted' targets (var genes with mismatches) bound by dCas9 should be included in Supp. Table S1 and in the alignment in Supp. Fig. S1.

Page 9. Profilin and tubulin beta chain, discussed in the text, should be indicated in fig 3a.

Page 10. The nomenclature of the domains should be consistent between the figure and the text (e.g. the S5 fold domain discussed in the text is labeled as MORC in fig 3).

In general, AP2 should be used to refer to the domain, and ApiAP2 to refer to the protein family.

Fig 1B, 'Both' instead of 'Either'. Here and elsewhere, IX instead of VIV.

Fig 2A, the other unintended /off target binding events described in the text should also be indicated by asterisks.

Fig 2B, provide gene IDs in full.

Supp. fig S2. The quality of the image should be improved.

Abstract. The authors should refer to the identification of novel factors, rather than 'de novo identification', because they don't identify new proteins in the sense of peptides originating from non-coding regions.

Reviewer #3:

This manuscript by Bryant et al, describes the use of dead Cas9 (dCas9) to mark and isolate proteins from var gene promoters and introns in the malaria-causing parasite *Plasmodium falciparum*. Var gene biology in *P. falciparum* is intellectually intriguing whilst also being important for the virulence of the deadliest form of malaria. Var genes are expressed in a mutually exclusive fashion and regulated by epigenetic mechanisms whereby all but one var genes are silenced and switching occurs between different members to allow for antigenic variation. Because var genes encode the major cytoadhesion protein PfEMP1, which binds vascular endothelial cells, this allows for sequestration of infected RBC in the face of an adaptive immune system, thus persistence of infection for ultimate transmission.

Despite the importance of var gene epigenetic regulation comparatively little is understood about how this process works, as compared to model systems. This manuscript takes an unbiased approach to find new factors in the regulation of var genes, by marking var promoters and introns with dCas9, then immunoprecipitating these regions to identify proteins involved in their regulation. Several chromatin factors were identified and an ISWI orthologue was focused on. A knockdown of ISWI was generated and implicated in the positive regulation of var gene expression.

Overall this manuscript is well put together, the data solid and contains important contributions to the field. I only have minor points:

1. Fig 1B: Why is H3 not enriched in the pulldown? This would be expected if dCas9 is able to associate with chromatin
2. Fig 2C: What do the values in the '['] mean? Why is it that there is incomplete coverage of the var transcripts in the RIP? Does dCas9 localisation affect transcription or translation of var genes? If this is altered it might affect the output.
3. Fig 3B: What type of phylogenetic tree has been constructed? What are the statistical values associated with the important branchpoints?
4. It is noted that the authors were unable to 'perform' gene knockout. Do you mean unable to 'achieve' a gene knockout?
5. Fig 4: The most significant GO term enrichment was 'ribosome'. This suggests that the knockdown is either having a level of pleiotropic effects or causing a delay in cell cycle progression. This should at least be acknowledged.

6. Fig 4C: Please label some more genes affected by the loss of iswi in this graph. Also, is there any position-effect of iswi-dependent gene expression or association with var gene loci? Why was IP-LC-MS/MS done on iswi and not ChIP? This would provide a clearer picture of the role of iswi in its role in var gene expression over a more general role.

An appeal was requested by the corresponding Author on behalf of all contributing Authors.

Thank you for your message asking us to reconsider our decision regarding your manuscript MSB-20-9569. I have now had a chance to read your preliminary point-by-point response and have also discussed it with the other members of the editorial team. Based on the outline you provide, we think that the proposed revisions sound reasonable. As such, we would invite you to submit a revised version of the current study, provided that the issues raised by the reviewers can be convincingly addressed, as detailed in my previous decision letter.

With regards to the comment 2 from reviewer #2, I would like to point out that the reviewer was not referring to the 50% reduction in iswi expression, but to the even smaller reduction in var expression, which has been confirmed by this reviewer.

On a more editorial level, please do the following.

Rebuttal letter to Reviewers comments:

Reviewer #1:

This manuscript uses an innovative approach, dCas9 targeting and immunoprecipitation, to attempt to identify proteins associated with the regulatory elements of var genes, the multi-gene family that are responsible for Plasmodium falciparum cytoadherence and therefore play a key role in malaria pathogenesis. Control of var gene expression is a significant area of focus for malaria biology and this novel piece of work identifies new potential components of what is undoubtedly a complex molecular system. The manuscript is well written and well referenced, with methods described in commendable detail, and data is being made broadly open access, also to be commended. Enthusiasm for the manuscript in its current form is however diminished by issues with control, quantification and validation, which make the key conclusions currently less than convincing. These need addressing or clarifying in a revised version.

Major points

1. dCas9 targeting control. Distinguishing between var-specific factors and general chromatin associated factors was surely one of the key goals of the work - that is certainly how the research goal is positioned in the Introduction. If that is the case, having a control line that helps to distinguish var-specific signal from general noise would seem to be key. The control line is however described as "a non-specific sgRNA that does not target any genomic sequence". It's not quite clear what this means, and I could not find any reference to the specific sgRNA sequence in the Methods (apologies if I overlooked it). Was the control sgRNA deliberately designed to not bind to any sequence in the P. falciparum genome? If so, how was this achieved bioinformatically? If not, how was it designed?

Reply:

The non-targeted guide RNA consists of a 34-nucleotide crRNA followed by an optimized tracrRNA, which is required for dCas9 binding. The crRNA is the sequence of the BtgZI cloning site in the optimized pL6_sgRNA plasmid: 'CCTAGGAACTCATCGCTCGCGATGCTGCCCGACA'. We have added this information to the Materials and Methods section and Table 1. With short read alignment using bowtie analysis, we observe that this sequence or the final 20 nucleotides adjacent to the tracrRNA do not correspond to any sequence in the P. falciparum 3D7 genome with three or less mismatches (i.e. three mutations would be required to obtain specificity for a sequence in the genome). Using bowtie2 and bwa with default settings for the entire sequence or the final 20 nucleotides adjacent to the tracrRNA, we observe no read mapping to the genome. We have added a circos plot of input-normalized chromatin immunoprecipitation and sequencing data for the non-targeted dCas9, which shows only background binding to the genome (Figure EV2A).

More importantly, what was the rationale for this? Surely to identify var-specific factors, the key control would be to target dCas9 to a different chromosomal location, not associated with var genes, but still actively transcribed during the 12-14 hour life cycle stage? Repeating the IP and mass spec with such a control would of course be a very large amount of work, and the lack of it

doesn't completely invalidate the findings, but at the very least, the rationale for why this approach was taken would be helpful, and a much clearer explanation of how and why the control sgRNA was chosen.

Reply:

By western blot, we see that dCas9 is not expressed at high levels; however, one must assume that the vast majority of dCas9 in the nucleus is not bound to the targeted locus.

The rationale of our approach of using a dCas9 binding to multiple var loci (~17) was to increase the signal-to-noise ratio of DNA-bound versus unbound dCas9. If a single locus (such as the promoter of the actin gene) would be targeted, only a single molecule of dCas9 would actually bind to the target, with the large majority of dCas9 still remaining unbound. Therefore, we believe that with regard to the proteomics control, a dCas9 targeted to no specific locus would give a similar level of non-specific chromatin interactions (background) as a dCas9 targeted to a single locus. We have added the following sentences to the Discussion to address this point: "Label-free quantitative proteomics and comparison of the promoter- and intron-targeted dCas9 immunoprecipitations to each other and to the non-targeted dCas9 immunoprecipitation allowed for the identification of proteins that were specific to each DNA element (Fig 3). Using a non-targeted dCas9 that does not bind specifically to the genome for normalization, we identified var gene specific factors such as SIP2 and other proteins that were shown to bind to or near var genes like the Alba proteins, HP1, and H2A.Z (Flueck et al, 2009; Pérez-Toledo et al, 2009; Petter et al, 2011; Chene et al, 2012; Goyal et al, 2012)."

With regard to identification of factors that are specific to var genes, we do identify SIP2, which was shown by a previous study to be specific for var genes (Flueck et al, 2010). Since we find SIP2 enriched at var gene promoters, we believe that our technique is able to identify factors that are specific to var genes and factors that are important for var gene regulation and involved in the general transcriptional machinery. However, the overall goal of this study was not to identify factors that are specific to var genes, but that are involved in var gene transcription or biology. We state this in the introduction: "Thus, a new, unbiased approach is needed to identify novel var gene-interacting factors that contribute to transcriptional regulation and organization of var genes." As we state throughout the manuscript, very few proteins in general have been shown to be involved in var gene regulation. Moreover, no protein factors have been shown to play a role in the activation of the single active var gene. Thus, we believe that the identification of any novel chromatin-associated proteins is interesting and significant. In other eukaryotes, HP1 was shown to be a general transcriptional repressor of facultative heterochromatin. However, in Plasmodium, HP1 was shown in 2009 to play an interesting role in repression of var genes, other clonally variant gene families, and ap2-g, which transformed the field of Plasmodium epigenetics. While ISWI may not be the master regulator of var gene mutually exclusive expression, we have identified it as an important piece of the puzzle, especially the black box that is var gene activation. The ISWI IP-mass spec offers a list of associated proteins that provide clues into how specificity might be achieved for various cohorts of genes, which we address in the discussion.

Because so little is known in general about transcription, DNA replication and repair, and telomere biology in Plasmodium, we believe that our study, which identified a new ISWI/MORC

complex that associates with var genes, does provide an important advancement in the field. Multiple studies (Toenhake et al., 2018, Ruiz et al., 2018) have recently shown that promoter and chromatin accessibility plays a major role in the transcriptional control program during intraerythrocytic development, and our approach identifies candidate proteins and perhaps complexes facilitating this phenomenon.

2. Western blot validation. On a similar issue, the Western blot shown in Fig 1C establishes that one of the targeted dCas9 constructs localises primarily to chromatin, but it is not specified which one (var-promoter or intron) and critically the control targeted dCas9 is not shown to establish whether it is even nuclear in localisation. Western blots with fractionated material is needed from all three lines to make it clear what is actually being compared in the subsequent ChIP-seq and mass spec experiments. If the control line does not have nuclear targeted dCas9, it is not an effective control for subsequent experiments.

Reply:

The western blot in the manuscript is for the promoter-targeted dCas9. We have clarified this in the figure legend.

The pUF-dCas9 plasmid is the same in all three strains, and the dCas9 is fused to a triple nuclear localization signal. This dCas9 expression system has been used in multiple published studies (Baumgarten et al., 2019; Barcons-Simon et al., 2020), and the localization of dCas9 to the nucleus is independent of sgRNA. We have added a western blot (Figure EV1D) that shows the fractionation of the non-targeted dCas9. dCas9 is present in the nuclear and chromatin fraction, but our dCas9 ChIP-sequencing data show that binding to the genome is random and non-specific (Figure EV2A).

In addition to these western blots, our proteomics analysis shows that dCas9 was successfully immunoprecipitated from the chromatin fraction of each replicate of each strain (see our response to the next question).

3. Mass spectrometry interpretation. It is not clear why mass spectrometry was not carried out for the control line. Given the number of cytoskeleton-associated proteins enriched with the intron targeted dCas9 in particular, there is obviously some concern about non-specific pulldown of abundant proteins. This is an issue with any mass spec study of course, which is why controls are so important. The increase of such abundant, potentially non-specific, proteins in the intron targeted line may be linked to the fact that the ChIP-seq data showed that binding of dCas9 was less enriched at the intron targeted sequences than the promoter targeted sequence, implying a higher level of background binding. This emphasises again that dCas9 targeted to a different gene would seem to have been the perfect control for the mass spectrometry study. While reluctant to ask for another large set of experiments, it is not clear without such data how the mass spectrometry findings can be interpreted clearly. Presumably also the approach of looking for proteins enriched at one or other location also overlooks the potential to identify proteins that are found at both, yet are still var-specific.

Reply:

In our initial analysis comparing the var gene intron- and promoter-targeted dCas9 samples, we sought to identify specific factors for each genomic region that, given their genomic distance, would serve as internal controls for each other (i.e. general chromatin-associated factors would not be identified as enriched in either sample). However, we agree with the reviewer and have performed the proteomics analysis with a non-targeted dCas9 control, which generally confirms and, in fact, improves our initial findings. We have incorporated the new data into the revised version of the manuscript (Figure 3, Figure EV3A, Tables EV4-10). Here, we provide a detailed explanation of our analysis:

For our study, label-free quantitative proteomics allowed us to analyze different samples with low amounts of input material using a sensitive mass spectrometer. This type of experiment allows one to quantify proteins and compute their ratios across samples, which is useful because a protein could be enriched in a test condition even though it is also present in a control condition. The comparison of samples (i.e. promoter-targeted versus non-targeted dCas9) relies on the assumption that the protein levels of the bait (dCas9) are the same in all replicates (four for each strain). However, despite using the same number of parasites and experimental conditions for each immunoprecipitation experiment and performing the mass spectrometry of all replicates for all samples on the same day, we detected more dCas9 in the non-targeted immunoprecipitation samples and significantly fewer dCas9 peptides in two replicates of the promoter-targeted dCas9 condition (Figure EV3A and Table EV4). The same significant trends can be seen for the dCas9 peptides ion intensity (Figure 1 below). We initially chose to perform only the promoter- versus intron-targeted dCas9 analysis because while the ratio of dCas9 ions was 1.27 ($p = 3 \times 10^{-21}$) in the intron/promoter-targeted dCas9 comparison, it was 3.16 ($p = 1.88 \times 10^{-62}$) in the non-targeted/intron-targeted comparison and 3.81 ($p = 1.21 \times 10^{-9}$) in the non-targeted/promoter-targeted comparison (Table EV4 bottom). A correlation can be seen between the number of dCas9 peptide ions detected and the total number of proteins detected by mass spectrometry (Table EV4 top and Figure EV3A). Thus, the non-targeted dCas9 is an extremely stringent control for label-free quantitation, and many proteins that are truly enriched in the promoter- or intron-targeted dCas9 will be lost with even a 1.5 fold enrichment cutoff when compared to the non-targeted dCas9.

The non-targeted control is even more stringent for the promoter-targeted dCas9, which had two replicates with significantly lower levels of dCas9. For label-free quantification, replicates are normalized based on total protein intensity distribution in order to compensate intensity variability between replicates and make them comparable between conditions. Two promoter-targeted dCas9 replicates had a median intensity below the other promoter-targeted replicates and especially all replicates of the intron- and non-targeted dCas9 samples (Figures 2 & 3 below). Even after normalization, these two promoter-targeted replicates have a different mean intensity than the other replicates (Figures 2 & 3 below, compare graphs on the left to those on the right). The same difference is not seen in the normalization between the intron- and non-targeted dCas9 replicates (Figure 4 below). Thus, in order to avoid experimental bias, we have chosen to exclude these two promoter-targeted replicates from the analysis. In doing so, we change the ratio of dCas9 ions in the non-targeted versus the promoter-targeted samples from

3.81 to 3.07. Without these two promoter-targeted replicates, the ratio of dCas9 ions in the promoter-targeted versus intron-targeted sample becomes 1.04.

Under these circumstances, we believe that excluding these two replicates is justified because it allows for a more unbiased analysis. Indeed, the new analysis identified more proteins that have previously been shown to be associated with var genes. In the new analysis of the promoter-targeted sample, we now find SIP2, HP1, and H2A.Z enriched in the promoter-targeted dCas9 immunoprecipitation (ratio ≥ 1.5 , $p \leq 0.05$) when compared to the intron-targeted (Table EV5) or non-targeted dCas9 samples (Table EV7). All of these proteins have been shown to be enriched in the upstream promoter regions of var genes (Flueck et al., 2009; Flueck et al, 2010; Petter et al., 2011). For the new comparison of the promoter-targeted dCas9 with the non-targeted dCas9 samples, all of the proteins we discussed in the previous analysis with the intron-targeted dCas9 (MORC, MDM2, CHD1, and ISWI), with the exception of SMC1, still show an enrichment in the promoter-targeted dCas9 samples (Table 1 below). Thus, we have removed SMC1 from the manuscript. In addition, we provide a Gene Ontology analysis of all significant promoter-enriched proteins shared between the intron- and non-targeted comparisons in the manuscript (Table EV10). Top GO enrichment categories include “chromatin assembly”, “chromosome organization”, and “DNA replication”. Thus, we believe we are justified in our focus on putative nucleosome remodelers. We have added a discussion of proteins involved in DNA replication and repair (DH60, CAF2, MCM2,4,6,7, RFC4, and MSH2,6), which we believe may be involved in mitotic recombination of var genes, considering that we harvested the parasites well before DNA replication. We have updated Figure 3A to reflect these changes and have added Figure 3B to compare the promoter/intron and promoter/non-targeted dCas9 comparisons. In addition, we have added Figures 3C and 3D to highlight the significant proteins that are shared between the ISWI, dCas9 promoter/intron, and dCas9 promoter/non-targeted immunoprecipitation analyses.

For the new intron-targeted dCas9 analysis (compared to two replicates of promoter-targeted dCas9 immunoprecipitation, Table EV6), we see a significant enrichment (ratio ≥ 1.5 , $p \leq 0.05$) of Alba proteins 1-3, as well as myosin A, coronin, and profilin. While we still see an enrichment of Alba 4, tubulin beta chain, and actin I, the ratios are not significant and we have removed these factors from the manuscript (Table 2 below). However, we have found actin-related protein 4 to be significantly enriched and have added this factor to the manuscript. While actin I, myosin A, tubulin beta chain, coronin and profilin are enriched in the intron-targeted dCas9 sample compared to the non-targeted dCas9 sample (Table EV8), only coronin and profilin show significant enrichment ratios (Table 2 below). We have updated Figure 3A to reflect these changes. We also provide a Gene Ontology analysis of all significant intron-enriched proteins shared between the promoter- and non-targeted comparisons in the manuscript (Table EV9), with two of the top GO enrichment categories being “localization of cell” and “cell motility”.

While some of these promoter- or intron-enriched factors have ratios less than 1.5 and p-values above 0.05 in the comparison with the non-targeted control, it must be kept in mind that the non-targeted dCas9 control has a ~3 fold higher amount of dCas9 than the promoter- and intron-targeted samples. Because we found that the number of dCas9 peptides correlates with the number of total proteins detected with LC-MS/MS (Figure EV3A), the non-targeted dCas9 is an overly stringent control. Even though ISWI does not show a statistically significant enrichment in

the promoter-targeted dCas9 compared to the non-targeted dCas9, we have shown that ISWI does bind to the var gene promoter with chromatin immunoprecipitation and sequencing. Moreover, ISWI IP LC-MS/MS identifies several proteins that are enriched in the promoter-targeted dCas9 samples compared to the intron- or non-targeted dCas9 samples. This suggests that we would be underestimating the number of proteins in the promoter- and intron-targeted dCas9 samples if we only compared them to the non-targeted dCas9 sample.

Figure 1. *dCas9 peptides ion intensity (log2) distribution between replicates of intron- (red), promoter- (green), and non-targeted (blue) dCas9. Grey line represents the global mean peptides ion intensity. The grey point in each boxplot represents that replicate's mean peptide ion intensity.*

Figure 2. Protein intensity (\log_2) distribution between replicates before (left graph) and after (right graph) normalization in promoter-targeted (blue) and non-targeted (red) dCas9 replicates. The two promoter-targeted dCas9 replicates with low intensity are circled.

Figure 3. Protein intensity (\log_2) distribution between replicates before (left graph) and after (right graph) normalization in promoter-targeted (red) and intron-targeted (blue) dCas9 replicates. The two promoter-targeted dCas9 replicates with low intensity are circled.

Figure 4. Protein intensity (\log_2) distribution between replicates before (left graph) and after (right graph) normalization in non-targeted (red) and intron-targeted (blue) dCas9 replicates.

Gene	Description	Old Ratio Intron	New Ratio Intron	Ratio Control	Old p-value Intron	New p-value Intron	p-value Control
PF3D7_0604100	SIP2	8.52	11.94	3.93	1.85E-02	1.63E-02	2.05E-02
PF3D7_1130700	SMC1	∞	----	----	----	----	----
PF3D7_1468100	MORC family protein, putative	1.61	2.13	1.40	2.61E-08	4.58E-11	1.16E-04
PF3D7_0518200	SWIB/MDM2 domain-containing protein	2.00	1.91	2.31	6.41E-04	9.67E-04	2.66E-02
PF3D7_1023900	CHD1	2.30	2.80	1.38	1.73E-08	1.93E-08	8.95E-04
PF3D7_0624600	ISWI	1.64	1.97	1.10	6.01E-22	1.02E-20	1.83E-01
PF3D7_1104200	SNF2L	1.44	1.87	1.15	1.10E-04	1.21E-05	5.19E-02
PF3D7_1220900	HP1	1.55	2.69	1.57	1.06E-01	2.43E-11	1.29E-02
PF3D7_0320900	H2A.Z	0.85	1.70	1.79	4.66E-01	1.02E-02	1.11E-03
PF3D7_1227100	DNA helicase 60	2.43	3.87	4.42	2.52E-01	1.87E-02	4.35E-03
PF3D7_1329300	CAF2	3.37	3.72	2.36	1.48E-02	1.82E-03	1.68E-02
PF3D7_1417800	MCM2	3.14	3.97	2.17	2.16E-09	9.91E-11	3.98E-10
PF3D7_1317100	MCM4	3.78	4.96	2.68	2.06E-15	1.57E-16	1.07E-12
PF3D7_1355100	MCM6	4.37	7.57	3.05	1.14E-09	4.76E-05	1.66E-05
PF3D7_0705400	MCM7	2.47	3.45	2.32	5.49E-09	1.56E-13	9.53E-08
PF3D7_1241700	RFC4	1.69	1.87	1.67	3.17E-05	2.64E-10	2.98E-03
PF3D7_1427500	MSH2	2.62	3.45	1.93	1.84E-11	1.44E-12	8.10E-06
PF3D7_0505500	MSH6	2.81	3.47	2.02	1.09E-17	1.51E-12	1.35E-06
	Cas9	0.73	1.04	0.33	1.20E-44	1.47E-01	5.64E-86

Table 1. Enrichment ratios and p-values for proteins found in the promoter-targeted dCas9 immunoprecipitation when compared to the intron-targeted or non-targeted (Control) dCas9 immunoprecipitations. “Old” refers to the original analysis using all four replicates of the promoter-targeted dCas9 immunoprecipitation, and “New” refers to the new analysis using only the two replicates having dCas9 and total protein levels comparable to the replicates in the intron- and non-targeted dCas9 samples. Ratios ≥ 1.5 are highlighted in green, ratios between 1 and 1.5 are highlighted in yellow, and ratios below 1 are highlighted in orange. p-values > 0.05 are highlighted in red.

Gene	Description	Old Ratio Promoter	New Ratio Promoter	Ratio Control	Old p-value Promoter	New p-value Promoter	p-value Control
PF3D7_1246200	actin I	1.96	1.30	1.34	2.69E-36	1.18E-07	2.30E-13
PF3D7_0814200	DNA/RNA-binding protein Alba 1	1.76	1.57	0.59	6.39E-08	6.60E-04	2.39E-17
PF3D7_1346300	DNA/RNA-binding protein Alba 2	2.37	1.52	0.84	3.29E-12	2.40E-05	6.04E-02
PF3D7_1006200	DNA/RNA-binding protein Alba 3	2.05	1.62	0.77	3.52E-09	4.22E-06	1.13E-04
PF3D7_1347500	DNA/RNA-binding protein Alba 4	1.73	1.28	0.79	5.92E-14	2.23E-07	4.44E-10
PF3D7_1342600	myosin A	1.97	1.58	1.46	2.45E-10	7.65E-05	2.11E-05
PF3D7_1251200	coronin	1.84	1.78	1.82	4.32E-05	4.76E-05	9.53E-06
PF3D7_0932200	profilin	1.86	1.52	1.82	4.96E-05	2.57E-02	1.48E-04
PF3D7_1008700	tubulin beta chain	2.03	1.49	1.09	1.40E-15	7.64E-08	1.77E-01
PF3D7_1422800	ARP4a	1.60	∞	0.82	4.43E-01	----	6.21E-01
	Cas9	1.36	0.96	0.32	1.20E-44	1.47E-01	1.88E-62

Table 2. Enrichment ratios and p-values for proteins found in the intron-targeted dCas9 immunoprecipitation when compared to the promoter-targeted or non-targeted (Control) dCas9 immunoprecipitations. “Old” refers to the original analysis using all four replicates of the promoter-targeted dCas9 immunoprecipitation, and “New” refers to the new analysis using only the two replicates having dCas9 and total protein levels comparable to the replicates in the intron- and non-targeted dCas9 samples. Ratios ≥ 1.5 are highlighted in green, ratios between 1 and 1.5 are highlighted in yellow, and ratios below 1 are highlighted in orange. p-values > 0.05 are highlighted in red.

4. Validation. ISWI is chosen as the sole target for validation, and immunoprecipitation of ISWI-3HA co-purifies several proteins that were also identified in the initial mass spec. This is encouraging, but it is also critical to establish whether ISWI-3HA localises to var promoters, either by CHIP-seq and/or an immunolocalization approach. This will distinguish between ISWI being a background interaction, or one that is specifically associated with var gene regulation, which is after all the goal of the project.

Reply:

We thank the reviewer for appreciating the ISWI immunoprecipitation-mass spec approach, which we see as an important verification of the initial dCas9 approach. We want to reiterate, however, that the overall goal of this study was not to identify factors that are specific to var genes, but that are involved in var gene transcription or biology.

To be completely honest, we have attempted to tag/knockout/knockdown at least a dozen of the significant hits from our study (for example SMC1, CHD1, MSH6, MORC, and CAF2) over the past year. However, we have only had success with the ISWI-HA knockdown strain. We have now successfully performed ChIP-seq for ISWI-HA and have included this data in Figure 4. We show that ISWI binds to the promoter of the active var gene, which validates our dCas9 IP LC-MS/MS approach (Figure 4G,H). The ChIP-seq data also show ISWI enrichment in the promoters of the genes that are down-regulated in response to ISWI knockdown (including the active var gene), the majority of which reach their highest transcriptional levels in ring stage (Figure 4C,D). These data suggest a direct effect of ISWI binding on transcriptional activation. While ISWI may not be specific to var genes, we believe that the association of ISWI, a putative chromatin remodeler, with accessible intergenic regions of the genome (Figure 4E) and with MORC, ACS, and an ApiAP2 transcription factor (Figure 3C,D) has important implications for regulation of var and other genes and will spawn an interesting new line of research in P. falciparum epigenetics. We have changed any language in the manuscript suggesting that ISWI is specific to var genes.

Minor points

1. Synchronisation. Tight synchronisation is obviously key to the comparability between replicates for the mass spec studies. While the synchronisation method is described, was synchrony actually quantitatively or even semi-quantitatively measured in each replicate, for example by qRT-PCR, or even simple morphology, to confirm that the replicates were broadly comparable?

Reply:

We routinely synchronize our parasites and always check for synchronicity with Giemsa staining, as we state in the first section of our Materials and Methods. At the time of harvest, we verified that all replicates were at the same ring stage. In addition, we have performed statistical estimation of cell cycle progression as in (Lemieux et al., 2009) by comparing RNA sequencing data from each strain to the microarray data in (Bozdech et al., 2003), in which gene transcription was measured at one-hour intervals over the course of intraerythrocytic development. We did not perform RNA sequencing for all four replicates for each strain, but the replicates we did analyze show that our synchronization methods yielded synchronous cultures that were at a time point in the cell cycle that most similarly corresponds to the 14 hpi time point in (Bozdech et al., 2003). We have added this data in Figure EV2C.

2. ISWI-3HA IP control. Why were wildtype parasites used as an IP control, which would presumably would not yield any immunoprecipitated protein at all, as there is no anti-HA target in the line? The lack of any immunoprecipitate will significantly reduce background, and also therefore significantly reduce the ability to distinguish signal from noise in the ISWI-3HA IP material. Another HA-tagged protein would be a much better control – given that they also have generated dCas9-3HA, surely this would have provided a better control for this IP, even without gRNA targeting?

Reply:

We believe that using a wild-type line that does not express an HA-tagged protein to account for non-specific protein binding to the antibody and beads is a valid and accepted control for IP-mass spec analyses. Indeed, we identified over 200 putatively background proteins in this control. We did perform an IP-mass spec analysis of another HA-tagged protein for an unrelated project in the lab in the same run with the ISWI-HA IP-mass spec. This other HA-tagged protein showed a completely different set of enriched proteins than ISWI-HA (unpublished data).

We did not use dCas9-HA as a control because it is a foreign protein and this strain is under drug selection to maintain the episome containing the dCas9 gene. We were trying to keep the conditions between the control and test strain as similar as possible.

Reviewer #2:

In this manuscript, Bryant et al. develop a method based on CRISPR-dCas9 targeting and quantitative proteomics to identify proteins associated in vivo with specific regions of the *Plasmodium falciparum* genome. This is a major technical development that will enable the identification of the malarial regulatory proteins associated with specific loci. The authors carefully demonstrate the specificity of the binding of dCas9 to the intended loci using ChIP-seq analysis. They use the method to characterize the proteins that bind the promoter and intron of var virulence genes, which results in the identification of a large number of differentially bound proteins. One of these proteins, the chromatin remodeler ISWI, is further characterized using knockdown and co-immunoprecipitation approaches.

The establishment of the CRISPR-based immunoprecipitation-proteomics method is a big advance for the field, but the manuscript provides limited biological insight on the regulation of var genes. Furthermore, there are important issues with the interpretation of some of the experiments and a key control is missing.

Major comments.

1. An important limitation of this study is that the transcriptional state of the targeted var genes has not been unambiguously determined, and consequently it is unclear whether the proteins identified associate with the chromatin of the active or the silenced state. If dCas9 binding does not alter the normal regulation of var genes, mutually exclusive expression implies that out of the 15-19 genes targeted, only one (or none in parasites in which a non-targeted var gene is expressed) is in an active state in each parasite.

Reply:

The objective of our study was to identify novel factors associated with putative var gene regulatory elements that we and others could further investigate with regard to var gene transcription and biology. We clearly state in the Discussion of the manuscript: "For our analysis, we exploited the homology of targeted sequences within the var gene family and maximized protein content of the immunoprecipitation by targeting dCas9 to multiple var genes

in the same strain with a single sgRNA (Figures 1 and 2). Thus, it is possible that we immunoprecipitated both silent and active var genes simultaneously from non-clonal bulk cultured parasites, as they collectively did not express a single var gene.” Even if we were able to repeat our experiment and target a single var gene known to be active or silent, we would presumably immunoprecipitate a subset of proteins that we have already identified with our current experimental approach. However, as the reviewers point out, the identified proteins would still need to be validated and characterized, which provides the only definitive evidence as to whether an identified factor is involved in activation or silencing. Thus, the downstream analysis would be the same and the biological insight would be gained from either approach only from targeted functional characterization, which we have done with ISWI.

Thus, the majority of factors identified are likely associated with the silenced state. However, the authors claim that in the promoter-targeted dCas9 line, the dCas9-bound var genes are expressed at higher level, based on RIP-seq and RNA-seq data. Unfortunately, the data is not shown in an informative way. Only two specific genes are shown in fig 2c (without showing the non-targeted dCas9 control strain for comparison), and in the supplementary tables, RNA-seq data is only shown for a subset of the genes with differential RIP-seq signal. Whether or not other var genes (those not bound by dCas9) are also expressed at higher levels in the promoter-targeted dCas9 strain is not described. The RNA-seq and RIP-seq data should be presented in a way that can be evaluated by the reader. The authors should provide, at least, complete supplementary datasets, and a bar chart including all targeted and non-targeted var genes (analogous to the chart in fig 4d).

Reply:

We have uploaded all raw RNA-seq datasets for each strain to the NCBI BioProject with accession PRJNA529754 and now provide RIP-seq values for every gene in the genome in Table EV3. In addition, we now provide bar graphs showing var gene transcription in bulk cultures of each strain and indicate which genes are targeted by the intron- or promoter-directed dCas9 (Figure EV2D).

The intron-, promoter-, and non-targeted dCas9 strains are all bulk cultures that have the same parent strain. However, this does not mean that they all will show the same var gene transcription profiles, as we often see var gene switching during the time it takes for parasites to emerge after transfection. We do not want to speculate about why the targeted var genes are more highly expressed in the promoter-targeted dCas9 strain than in the non-targeted control strain. It is possible that this is just coincidence, as two bulk populations of parasites are likely to express different cohorts of var genes.

For the former Figure 2C, the promoter-targeted dCas9 RIP-seq data were normalized to the non-targeted dCas9 RIP-seq data. We have changed Figure 2C to show both tracks separately to make the point clearer. To give a genome-wide view of the data (rather than just two specific genes), we have added Figure EV2E, which is a circos plot providing a side-by-side comparison of promoter-targeted dCas9 ChIP (red) and RIP (green) sequencing data. We believe this circos plot shows quite clearly that dCas9 immunoprecipitates both DNA and associated RNA from the promoter-targeted var genes.

The suggestion by the authors that dCas9 binds to active promoters is not well-supported: even if dCas9 binds preferentially to active promoters, this wouldn't result in increased expression of the target genes as determined by RNA-seq if dCas9 binding does not alter expression. If, instead, the authors want to suggest that dCas9 binding results in increased expression of the target genes and disrupts mutually exclusive expression, such that the majority of dCas9-bound var genes are simultaneously active, this would need to be supported by clearly presented data, and it may represent an important concern for the interpretation of the data obtained with this system.

Reply:

We apologize for the misunderstanding, but with the text concerning this point in the results section, we are not trying to suggest in any way that dCas9 binding to a promoter results in increased expression of the target gene or preferentially binds to active var genes.

The purpose of the RIP-seq and the RNA-seq was to show that dCas9 binding to the targeted upstream region of the var gene does not prevent the targeted var gene's transcription, as has been seen for dCas9-mediated CRISPRi. Parallel dCas9 ChIP-seq and RIP-seq from the promoter-targeted strain shows that dCas9 binds to the targeted upstream DNA region of var gene X and is able to immunoprecipitate nascent RNA from the targeted var gene X, meaning that the targeted var gene is transcribed (Figure EV2E).

We have added a graph (Figure EV2B) that shows RNA sequencing data from clones of the intron- and promoter- targeted dCas9 strains. These data show that each clone transcribes a predominant var gene and maintains mutually exclusive transcription of the var gene family. We have deposited these RNA-seq datasets in the NCBI Bioproject database (PRJNA529754).

We have changed and expanded the paragraph in the Results section concerning the RNA- and RIP-seq from the dCas9 strains, to address the concerns of the reviewer.

There are specific systems that enable the study of var promoters in a controlled active or silenced state. The authors should investigate the effect of dCas9 binding on var gene expression in clones expressing a single specific var gene, or in transgenic lines in which a var promoter controls the expression of a drug resistance marker. Identification of proteins bound to var control elements at stages when the genes are not expressed (e.g. schizont stage) may also identify proteins specifically associated with the silenced state.

Reply:

Because we wanted to capture var genes in the most natural chromatin context possible, we avoided the use of strains in which a single var gene is activated with a drug resistance marker. It is unknown if such systems use the same epigenetic pathways to activate the manipulated var promoter.

We agree that performing our proteomics experiment with a single active or silent locus or at a later time point would be extremely interesting, and we would like to attempt this in the future. However, these experiments (especially single locus) are extremely complex and expensive, requiring a large amount of donated blood to culture and synchronize the number of parasites required. At this time, we do not have the resources or type of equipment (fermenter) required to obtain the number of parasites that would be needed for a single locus experiment.

2. The role of ISWI as a specific activator of var genes is not well demonstrated. First, the magnitude of the reduction of expression in the knockdown line is modest and may be explained by delayed life cycle progression as a consequence of glucosamine addition, or as a consequence of the reduced expression of ISWI. These possibilities should be excluded with appropriate control experiments.

Reply:

A 30% reduction in var gene transcription may seem modest, but it is statistically significant. Moreover, this reduction in transcription of the active var gene must be considered in light of the only 50% reduction we achieve in ISWI itself. We believe that the differential expression of genes is due to ISWI knockdown and is not a result of a delay in cell cycle progression based on the following data:

We routinely synchronize our parasites and always check for synchronicity with Giemsa staining. At the time of harvest, we saw no morphological differences between the control and glucosamine-treated replicates. We have performed a growth curve with the clone that we used for the differential expression analysis, which demonstrates that there is no difference in growth between control and glucosamine-treated parasites over the course of five days. We have added this graph as Figure EV4A and corresponding text to the manuscript.

In addition, we have performed statistical estimation of cell cycle progression as in (Lemieux et al., 2009) by comparing our RNA sequencing data to the microarray data in (Bozdech et al., 2003), in which gene transcription was measured at one-hour intervals over the course of intraerythrocytic development. All replicates used for our differential expression analysis (untreated and glucosamine-treated) were highly synchronous and were at a time point in the cell cycle that most similarly corresponds to the 12 hpi time point in (Bozdech et al., 2003). We have added this graph as Figure EV4B and corresponding text to the manuscript.

Second, over 100 genes show larger changes in expression than the var gene upon ISWI knockdown. Therefore, unless additional evidence is provided, ISWI appears to be a likely constitutive component of the basal transcriptional machinery, rather than a specific regulator of var gene expression. The argument in the Discussion (page 14) that if ISWI was required for general processes a more dramatic phenotype would be expected is not valid, because this may depend on the level of downregulation (which, according to fig 4a, is modest).

Reply:

We agree that ISWI itself is not specific to var gene transcription, and we have removed the sentence mentioned by the reviewer from the discussion. Indeed, we now show with ChIP-seq that ISWI enrichment in the promoter of a gene generally correlates with that gene's transcript levels.

The authors would like to make it clear that the overall goal of this study was not to identify factors that are specific to var genes, but that are involved in var gene transcription or biology. We state this in the introduction: "Thus, a new, unbiased approach is needed to identify novel var gene-interacting factors that contribute to transcriptional regulation and organization of var genes." As we state throughout the manuscript, very few proteins in general have been shown to be involved in var gene regulation. Moreover, no protein factors have been shown to play a role in the activation of the single active var gene. Thus, we believe that the characterization of any novel chromatin-associated proteins is interesting and significant. In other eukaryotes, HP1 was shown to be a general transcriptional repressor of facultative heterochromatin. However, in Plasmodium, HP1 was shown in 2009 to play an interesting role in repression of var genes, other clonally variant gene families, and ap2-g, which transformed the field of Plasmodium epigenetics. While ISWI may not be the master regulator of var gene mutually exclusive expression, we have identified it as an important piece of the puzzle, especially the black box that is var gene activation. The ISWI IP-mass spec offers a list of associated proteins (such as the ApiAP2 transcription factor highlighted in Figure 3D that is also enriched in the promoter-targeted dCas9 IP) that provide clues into how specificity might be achieved for various cohorts of genes. Because so little is known in general about transcription, DNA replication and repair, and telomere biology in Plasmodium, we believe that our study, which identified a new ISWI complex that associates with var genes, does provide an important advancement in the field. Multiple studies (Toenhake et al., 2018, Ruiz et al., 2018) have recently shown that promoter and chromatin accessibility plays a major role in the transcriptional control program during intraerythrocytic development, and our approach identifies candidate proteins and perhaps complexes facilitating this phenomenon.

3. My other major concern is about the quantitative proteomics-based identification of factors bound to the promoter or the intron. Unfortunately, control experiments with the non-targeted dCas9 strain were not performed. This control would be important to distinguish which of the many proteins identified are specifically bound to var genes, and which were non-specifically immunoprecipitated. Without this control, it is impossible to determine which of the >1,000 proteins actually participate in var regulation. The current analysis is focused on differential binding between promoter and intron, but this cannot replace an appropriate negative control with a non-targeted dCas9 strain. Furthermore, factors that bind to both the promoter and the intron may also be of interest. The analysis is selective, as it focuses on proteins involved in chromatin remodeling without even demonstrating that this functional category is enriched among the proteins identified, and many other highly enriched proteins are not discussed. An unbiased analysis of the data, in addition to the focus on specific candidate regulators, should be presented.

Reply:

In our initial analysis comparing the var gene intron- and promoter-targeted dCas9 samples, we sought to identify specific factors for each genomic region that, given their genomic distance, would serve as internal controls for each other (i.e. general chromatin-associated factors would not be identified as enriched in either sample). However, we agree with the reviewer and have performed the proteomics analysis with a non-targeted dCas9 control, which generally confirms and, in fact, improves our initial findings. We have incorporated the new data into the revised version of the manuscript (Figure 3, Figure EV3A, Tables EV4-10). Here, we provide a detailed explanation of our analysis:

For our study, label-free quantitative proteomics allowed us to analyze different samples with low amounts of input material using a sensitive mass spectrometer. This type of experiment allows one to quantify proteins and compute their ratios across samples, which is useful because a protein could be enriched in a test condition even though it is also present in a control condition. The comparison of samples (i.e. promoter-targeted versus non-targeted dCas9) relies on the assumption that the protein levels of the bait (dCas9) are the same in all replicates (four for each strain). However, despite using the same number of parasites and experimental conditions for each immunoprecipitation experiment and performing the mass spectrometry of all replicates for all samples on the same day, we detected more dCas9 in the non-targeted immunoprecipitation samples and significantly fewer dCas9 peptides in two replicates of the promoter-targeted dCas9 condition (Figure EV3A and Table EV4). The same significant trends can be seen for the dCas9 peptides ion intensity (Figure 1 below). We initially chose to perform only the promoter- versus intron-targeted dCas9 analysis because while the ratio of dCas9 ions was 1.27 ($p = 3 \times 10^{-21}$) in the intron/promoter-targeted dCas9 comparison, it was 3.16 ($p = 1.88 \times 10^{-62}$) in the non-targeted/intron-targeted comparison and 3.81 ($p = 1.21 \times 10^{-9}$) in the non-targeted/promoter-targeted comparison (Table EV4 bottom). A correlation can be seen between the number of dCas9 peptide ions detected and the total number of proteins detected by mass spectrometry (Table EV4 top and Figure EV3A). Thus, the non-targeted dCas9 is an extremely stringent control for label-free quantitation, and many proteins that are truly enriched in the promoter- or intron-targeted dCas9 will be lost with even a 1.5 fold enrichment cutoff when compared to the non-targeted dCas9.

The non-targeted control is even more stringent for the promoter-targeted dCas9, which had two replicates with significantly lower levels of dCas9. For label-free quantification, replicates are normalized based on total protein intensity distribution in order to compensate intensity variability between replicates and make them comparable between conditions. Two promoter-targeted dCas9 replicates had a median intensity below the other promoter-targeted replicates and especially all replicates of the intron- and non-targeted dCas9 samples (Figures 2 & 3 below). Even after normalization, these two promoter-targeted replicates have a different mean intensity than the other replicates (Figures 2 & 3 below, compare graphs on the left to those on the right). The same difference is not seen in the normalization between the intron- and non-targeted dCas9 replicates (Figure 4 below). Thus, in order to avoid experimental bias, we have chosen to exclude these two promoter-targeted replicates from the analysis. In doing so, we change the ratio of dCas9 ions in the non-targeted versus the promoter-targeted samples from 3.81 to 3.07. Without these two promoter-targeted replicates, the ratio of dCas9 ions in the promoter-targeted versus intron-targeted sample becomes 1.04.

Under these circumstances, we believe that excluding these two replicates is justified because it allows for a more unbiased analysis. Indeed, the new analysis identified more proteins that have previously been shown to be associated with var genes. In the new analysis of the promoter-targeted sample, we now find SIP2, HP1, and H2A.Z enriched in the promoter-targeted dCas9 immunoprecipitation (ratio ≥ 1.5 , $p \leq 0.05$) when compared to the intron-targeted (Table EV5) or non-targeted dCas9 samples (Table EV7). All of these proteins have been shown to be enriched in the upstream promoter regions of var genes (Flueck et al., 2009; Flueck et al, 2010; Petter et al., 2011). For the new comparison of the promoter-targeted dCas9 with the non-targeted dCas9 samples, all of the proteins we discussed in the previous analysis with the intron-targeted dCas9 (MORC, MDM2, CHD1, and ISWI), with the exception of SMC1, still show an enrichment in the promoter-targeted dCas9 samples (Table 1 below). Thus, we have removed SMC1 from the manuscript. In addition, we provide a Gene Ontology analysis of all significant promoter-enriched proteins shared between the intron- and non-targeted comparisons in the manuscript (Table EV10). Top GO enrichment categories include “chromatin assembly”, “chromosome organization”, and “DNA replication”. Thus, we believe we are justified in our focus on putative nucleosome remodelers. We have added a discussion of proteins involved in DNA replication and repair (DH60, CAF2, MCM2,4,6,7, RFC4, and MSH2,6), which we believe may be involved in mitotic recombination of var genes, considering that we harvested the parasites well before DNA replication. We have updated Figure 3A to reflect these changes and have added Figure 3B to compare the promoter/intron and promoter/non-targeted dCas9 comparisons. In addition, we have added Figures 3C and 3D to highlight the significant proteins that are shared between the ISWI, dCas9 promoter/intron, and dCas9 promoter/non-targeted immunoprecipitation analyses.

For the new intron-targeted dCas9 analysis (compared to two replicates of promoter-targeted dCas9 immunoprecipitation, Table EV6), we see a significant enrichment (ratio ≥ 1.5 , $p \leq 0.05$) of Alba proteins 1-3, as well as myosin A, coronin, and profilin. While we still see an enrichment of Alba 4, tubulin beta chain, and actin I, the ratios are not significant and we have removed these factors from the manuscript (Table 2 below). However, we have found actin-related protein 4 to be significantly enriched and have added this factor to the manuscript. While actin I, myosin A, tubulin beta chain, coronin and profilin are enriched in the intron-targeted dCas9 sample compared to the non-targeted dCas9 sample (Table EV8), only coronin and profilin show significant enrichment ratios (Table 2 below). We have updated Figure 3A to reflect these changes. We also provide a Gene Ontology analysis of all significant intron-enriched proteins shared between the promoter- and non-targeted comparisons in the manuscript (Table EV9), with two of the top GO enrichment categories being “localization of cell” and “cell motility”.

While some of these promoter- or intron-enriched factors have ratios less than 1.5 and p-values above 0.05 in the comparison with the non-targeted control, it must be kept in mind that the non-targeted dCas9 control has a ~3 fold higher amount of dCas9 than the promoter- and intron-targeted samples. Because we found that the number of dCas9 peptides correlates with the number of total proteins detected with LC-MS/MS (Figure EV3A), the non-targeted dCas9 is an overly stringent control. Even though ISWI does not show a statistically significant enrichment in the promoter-targeted dCas9 compared to the non-targeted dCas9, we have shown that ISWI does bind to the var gene promoter with chromatin immunoprecipitation and sequencing. Moreover, ISWI IP LC-MS/MS identifies several proteins that are enriched in the promoter-

targeted dCas9 samples compared to the intron- or non-targeted dCas9 samples. This suggests that we would be underestimating the number of proteins in the promoter- and intron-targeted dCas9 samples if we only compared them to the non-targeted dCas9 sample.

Figure 1. dCas9 peptides ion intensity (\log_2) distribution between replicates of intron- (red), promoter- (green), and non-targeted (blue) dCas9. Grey line represents the global mean peptides ion intensity. The grey point in each boxplot represents that replicate's mean peptide ion intensity.

Figure 2. Protein intensity (\log_2) distribution between replicates before (left graph) and after (right graph) normalization in promoter-targeted (blue) and non-targeted (red) dCas9 replicates. The two promoter-targeted dCas9 replicates with low intensity are circled.

Figure 3. Protein intensity (\log_2) distribution between replicates before (left graph) and after (right graph) normalization in promoter-targeted (red) and intron-targeted (blue) dCas9 replicates. The two promoter-targeted dCas9 replicates with low intensity are circled.

Figure 4. Protein intensity (\log_2) distribution between replicates before (left graph) and after (right graph) normalization in non-targeted (red) and intron-targeted (blue) dCas9 replicates.

Gene	Description	Old Ratio Intron	New Ratio Intron	Ratio Control	Old p-value Intron	New p-value Intron	p-value Control
PF3D7_0604100	SIP2	8.52	11.94	3.93	1.85E-02	1.63E-02	2.05E-02
PF3D7_1130700	SMC1	∞	----	----	----	----	----
PF3D7_1468100	MORC family protein, putative	1.61	2.13	1.40	2.61E-08	4.58E-11	1.16E-04
PF3D7_0518200	SWIB/MDM2 domain-containing protein	2.00	1.91	2.31	6.41E-04	9.67E-04	2.66E-02
PF3D7_1023900	CHD1	2.30	2.80	1.38	1.73E-08	1.93E-08	8.95E-04
PF3D7_0624600	ISWI	1.64	1.97	1.10	6.01E-22	1.02E-20	1.83E-01
PF3D7_1104200	SNF2L	1.44	1.87	1.15	1.10E-04	1.21E-05	5.19E-02
PF3D7_1220900	HP1	1.55	2.69	1.57	1.06E-01	2.43E-11	1.29E-02
PF3D7_0320900	H2A.Z	0.85	1.70	1.79	4.66E-01	1.02E-02	1.11E-03
PF3D7_1227100	DNA helicase 60	2.43	3.87	4.42	2.52E-01	1.87E-02	4.35E-03
PF3D7_1329300	CAF2	3.37	3.72	2.36	1.48E-02	1.82E-03	1.68E-02
PF3D7_1417800	MCM2	3.14	3.97	2.17	2.16E-09	9.91E-11	3.98E-10
PF3D7_1317100	MCM4	3.78	4.96	2.68	2.06E-15	1.57E-16	1.07E-12
PF3D7_1355100	MCM6	4.37	7.57	3.05	1.14E-09	4.76E-05	1.66E-05
PF3D7_0705400	MCM7	2.47	3.45	2.32	5.49E-09	1.56E-13	9.53E-08
PF3D7_1241700	RFC4	1.69	1.87	1.67	3.17E-05	2.64E-10	2.98E-03
PF3D7_1427500	MSH2	2.62	3.45	1.93	1.84E-11	1.44E-12	8.10E-06
PF3D7_0505500	MSH6	2.81	3.47	2.02	1.09E-17	1.51E-12	1.35E-06
	Cas9	0.73	1.04	0.33	1.20E-44	1.47E-01	5.64E-86

Table 1. Enrichment ratios and p-values for proteins found in the promoter-targeted dCas9 immunoprecipitation when compared to the intron-targeted or non-targeted (Control) dCas9 immunoprecipitations. “Old” refers to the original analysis using all four replicates of the promoter-targeted dCas9 immunoprecipitation, and “New” refers to the new analysis using only the two replicates having dCas9 and total protein levels comparable to the replicates in the intron- and non-targeted dCas9 samples. Ratios ≥ 1.5 are highlighted in green, ratios between 1 and 1.5 are highlighted in yellow, and ratios below 1 are highlighted in orange. p-values > 0.05 are highlighted in red.

Gene	Description	Old Ratio Promoter	New Ratio Promoter	Ratio Control	Old p-value Promoter	New p-value Promoter	p-value Control
PF3D7_1246200	actin I	1.96	1.30	1.34	2.69E-36	1.18E-07	2.30E-13
PF3D7_0814200	DNA/RNA-binding protein Alba 1	1.76	1.57	0.59	6.39E-08	6.60E-04	2.39E-17
PF3D7_1346300	DNA/RNA-binding protein Alba 2	2.37	1.52	0.84	3.29E-12	2.40E-05	6.04E-02
PF3D7_1006200	DNA/RNA-binding protein Alba 3	2.05	1.62	0.77	3.52E-09	4.22E-06	1.13E-04
PF3D7_1347500	DNA/RNA-binding protein Alba 4	1.73	1.28	0.79	5.92E-14	2.23E-07	4.44E-10
PF3D7_1342600	myosin A	1.97	1.58	1.46	2.45E-10	7.65E-05	2.11E-05
PF3D7_1251200	coronin	1.84	1.78	1.82	4.32E-05	4.76E-05	9.53E-06
PF3D7_0932200	profilin	1.86	1.52	1.82	4.96E-05	2.57E-02	1.48E-04
PF3D7_1008700	tubulin beta chain	2.03	1.49	1.09	1.40E-15	7.64E-08	1.77E-01
PF3D7_1422800	ARP4a	1.60	∞	0.82	4.43E-01	----	6.21E-01
	Cas9	1.36	0.96	0.32	1.20E-44	1.47E-01	1.88E-62

Table 2. Enrichment ratios and p-values for proteins found in the intron-targeted dCas9 immunoprecipitation when compared to the promoter-targeted or non-targeted (Control) dCas9 immunoprecipitations. “Old” refers to the original analysis using all four replicates of the promoter-targeted dCas9 immunoprecipitation, and “New” refers to the new analysis using only the two replicates having dCas9 and total protein levels comparable to the replicates in the intron- and non-targeted dCas9 samples. Ratios ≥ 1.5 are highlighted in green, ratios between 1 and 1.5 are highlighted in yellow, and ratios below 1 are highlighted in orange. p-values > 0.05 are highlighted in red.

Minor comments.

Having generated the transgenic line expressing HA-glms-tagged ISWI, the genome-wide distribution of tagged ISWI could be analyzed by ChIP-seq. This would help to clarify whether ISWI plays a preferential role in var regulation or is a general factor necessary for transcription of most genes.

Reply:

We have now successfully performed ChIP-seq for ISWI-HA and have included this data in Figure 4. We show that ISWI binds to the promoter of the active var gene, which validates our dCas9 IP LC-MS/MS approach (Figure 4G,H). The ChIP-seq data also show ISWI enrichment in the promoters of the genes that are down-regulated in response to ISWI knockdown (including the active var gene), the majority of which reach their highest transcriptional levels in ring stage (Figure 4C,D). These data suggest a direct effect of ISWI binding on transcriptional activation.

While ISWI may not be specific to var genes, we believe that the association of ISWI, a putative chromatin remodeler, with accessible intergenic regions of the genome (Figure 4E) and with MORC, ACS, and an ApiAP2 transcription factor (Figure 3C,D) has important implications for regulation of var and other genes and will spawn an interesting new line of research in P. falciparum epigenetics. We have changed any language in the manuscript suggesting that ISWI is specific to var genes.

Please provide a further analysis of the genes differentially expressed genes in the ISWI knockdown line. Are they among the most highly expressed genes at the ring stage? Are they genes with peak expression at other stages?

Reply:

We have performed the suggested analysis by comparing our RNA sequencing data to the data from (Bozdech et al., 2003), in which gene transcription was measured at one-hour intervals over the course of intraerythrocytic development. The majority of genes that are significantly down-regulated in response to ISWI knockdown normally reach peak expression in ring stage; however, genes that are up-regulated in response to ISWI knockdown normally reach their peak expression at later stages. These data suggest that ISWI plays a role in gene activation in ring stages and that the gene up-regulation seen with ISWI knockdown may be the result of pleiotropic effects. We have added a frequency plot as Figure 4C and corresponding text in the manuscript. In addition, we have added a meta-gene plot of ISWI ChIP-seq enrichment at down- and up-regulated genes as Figure 4D, which shows higher enrichment in the promoters of down-regulated genes.

Page 6. TSS is commonly used to refer to Transcription Start Sites. Using it to refer to the translation start site (ATG start codon) can be misleading. The scheme in Fig. 1A should reflect that promoter-targeted dCas9 is closer to the start codon than to the transcription start site (40 bp from the former and up to 2 Kb from the latter).

Reply:

We thank the reviewer for pointing this out and have modified Figure 1A to better represent the distances of dCas9 binding sites from the TSS and ATG.

Page 7. The off-target genes and 'unpredicted' targets (var genes with mismatches) bound by dCas9 should be included in Supp. Table S1 and in the alignment in Supp. Fig. S1.

Reply:

These unpredicted target sites were identified by dCas9 ChIP-sequencing, which gives a peak of dCas9 enrichment over 100 basepairs or more of DNA. For these unpredicted sequences, we are suggesting that binding is due to DNA motifs that are very similar to the intended target sequence (one mismatch), but we cannot prove this. However, we have added these sequences to the alignment in Figure EV1A-C and corresponding information to Tables EV1 (promoter) and EV2 (intron).

Page 9. Profilin and tubulin beta chain, discussed in the text, should be indicated in fig 3a.

Reply:

We have modified Figure 3A significantly, but have highlighted profilin in Figure 3A and 3B.

Page 10. The nomenclature of the domains should be consistent between the figure and the text (e.g. the S5 fold domain discussed in the text is labeled as MORC in fig 3).

Reply:

“MORC” has been changed to “S5”, but is now Figure EV3E.

In general, AP2 should be used to refer to the domain, and ApiAP2 to refer to the protein family.

Reply:

This has been corrected throughout the text.

Fig 1B, 'Both' instead of 'Either'. Here and elsewhere, IX instead of VIV.

Reply:

We believe that using “Both” would confuse readers and lead them to believe that we targeted the promoter and the intron at the same time. We have made the other suggested change, thank you for pointing this out.

Fig 2A, the other unintended /off target binding events described in the text should also be indicated by asterisks.

Reply:

We have added asterisks to PF3D7_0500100 and PF3D7_1300100 in the intron-targeted dCas9 ChIP-seq circos plot.

Fig 2B, provide gene IDs in full.

Reply:

This has been corrected.

Supp. fig S2. The quality of the image should be improved.

Reply:

This is a PDF generated by the Agilent 2100 Bioanalyzer, and we cannot modify it.

Abstract. The authors should refer to the identification of novel factors, rather than 'de novo identification', because they don't identify new proteins in the sense of peptides originating from non-coding regions.

Reply:

This has been changed.

Reviewer #3:

This manuscript by Bryant et al, describes the use of dead Cas9 (dCas9) to mark and isolate proteins from var gene promoters and introns in the malaria-causing parasite *Plasmodium falciparum*. Var gene biology in *P. falciparum* is intellectually intriguing whilst also being important for the virulence of the deadliest form of malaria. Var genes are expressed in a mutually exclusive fashion and regulated by epigenetic mechanisms whereby all but one var genes are silenced and switching occurs between different members to allow for antigenic variation. Because var genes encode the major cytoadhesion protein PfEMP1, which binds vascular endothelial cells, this allows for sequestration of infected RBC in the face of an adaptive immune system, thus persistence of infection for ultimate transmission.

Despite the importance of var gene epigenetic regulation comparatively little is understood about how this process works, as compared to model systems. This manuscript takes an unbiased approach to find new factors in the regulation of var genes, by marking var promoters and introns with dCas9, then immunoprecipitating these regions to identify proteins involved in their regulation. Several chromatin factors were identified and an ISWI orthologue was focused on. A knockdown of ISWI was generated and implicated in the positive regulation of var gene expression.

Overall this manuscript is well put together, the data solid and contains important contributions to the field. I only have minor points:

1. Fig 1B: Why is H3 not enriched in the pulldown? This would be expected if dCas9 is able to associate with chromatin

Reply:

While H3 does not appear in the western blot, it does appear in the proteomics analysis. We consistently see a decrease in the H3 levels in the supernatant of the immunoprecipitation by western blot, but the washes are perhaps stringent enough to remove most of the immunoprecipitated proteins except for the dCas9.

2. Fig 2C: What do the values in the '[']' mean? Why is it that there is incomplete coverage of the var transcripts in the RIP? Does dCas9 localisation affect transcription or translation of var genes? If this is altered it might affect the output.

Reply:

The bracketed values are:

- *ChIP: normalized coverage (per one million mapped reads) of HA ChIP minus input, as calculated with deeptool's 'bamCompare' (please see Materials and Methods)*
- *RIP: normalized coverage (per one million mapped reads) of HA RIP minus IgG RIP, as calculated with deeptool's 'bamCompare' (please see Materials and Methods)*
- *RNA-seq: normalized coverage (per one million mapped reads) over windows of one nucleotide*

Although we have now changed Figure 2C in order to address Reviewer 2's concerns, we would like to address the incomplete coverage observed in the RIP. Reduced coverage can be explained by the stringent mapping parameters we use, which allows a sequenced read to only map once back to the genome. Since the var gene family shows substantial homology, it sometimes leads to exclusion of sequences that are similar or the same amongst different var genes.

We do not believe that var gene translation is affected by dCas9 localization, but we have not performed the western blot analysis.

We cannot say to what extent dCas9 binding to the promoter effects var gene transcription, but we can say that dCas9 binding to the promoter does not block var gene transcription. Parallel dCas9 ChIP-seq and RIP-seq from the promoter-targeted strain shows that dCas9 binds to the targeted upstream DNA region of var gene X and is able to immunoprecipitate nascent RNA from the targeted var gene X, meaning that the targeted var gene is transcribed (Figure EV2E). RNA sequencing data from clones of both the intron- and promoter- targeted dCas9 strains show that each strain transcribes a predominant var gene and that all targeted var genes are not simultaneously active. These data have been added as Figure EV2B and are discussed in more detail in the Results section.

3. Fig 3B: What type of phylogenetic tree has been constructed? What are the statistical values associated with the important branchpoints?

Reply:

We constructed a maximum likelihood tree using the 'Le Guecal' (LG) model and 1,000 bootstrap replicates. We have added this information to the Material and Methods. The respective bootstrap values as calculated by MEGA v7 have been added on the branchpoints of the tree, which is now in Figure EV3D.

4. It is noted that the authors were unable to 'perform' gene knockout. Do you mean unable to 'achieve' a gene knockout?

Reply:

Thank you for pointing this out. We have changed the wording in the text.

5. Fig 4: The most significant GO term enrichment was 'ribosome'. This suggests that the knockdown is either having a level of pleiotropic effects or causing a delay in cell cycle progression. This should at least be acknowledged.

Reply:

We do not believe that the changes we see in the differential expression analysis are due to a delay in the cell cycle. We have performed a growth curve with the clone that we used for the differential expression analysis, which demonstrates that there is no difference in growth between control and glucosamine-treated parasites over the course of five days. We have added this graph as Figure EV4A and corresponding text to the manuscript. In addition, we have performed statistical estimation of cell cycle progression as in (Lemieux et al., 2009) by comparing our RNA sequencing data to the microarray data in (Bozdech et al., 2003), in which gene transcription was measured at one-hour intervals over the course of intraerythrocytic development. All replicates used for our differential expression analysis (untreated and glucosamine-treated) were highly synchronous and were at a time point in the cell cycle that most similarly corresponds to the 12 hpi time point in (Bozdech et al., 2003). We have added this graph as Figure EV4B and corresponding text to the manuscript.

While it is possible that the knockdown does have pleiotropic effects, we believe that these might be more represented by the up-regulated genes in the differential expression analysis (please see our response to your Question 6 below).

6. Fig 4C: Please label some more genes affected by the loss of iswi in this graph. Also, is there any position-effect of iswi-dependent gene expression or association with var gene loci? Why was IP-LC-MS/MS done on iswi and not ChIP? This would provide a clearer picture of the role of iswi in its role in var gene expression over a more general role.

Reply:

Given the size of the (now) Figure 4B and the number of genes that are significantly differentially expressed, it would be difficult to choose which genes to highlight in addition to ISWI. We hope that the following additional data will serve as an acceptable alternative:

First, we have compared our RNA sequencing data to the data from (Bozdech et al., 2003), in which gene transcription was measured at one-hour intervals over the course of intraerythrocytic development. The majority of genes that are significantly down-regulated in response to ISWI knockdown normally reach peak expression in ring stage; however, genes that are up-regulated in response to ISWI knockdown normally reach their peak expression at later stages. These data suggest that ISWI plays a role in gene activation in ring stages and that the gene up-regulation seen with ISWI knockdown may be the result of pleiotropic effects. We have added a frequency plot as Figure 4C and corresponding text in the manuscript.

Second, we have now successfully performed ChIP-seq for ISWI-HA and have included this data in Figure 4. We show that ISWI binds to the promoter of the active var gene, which validates our

dCas9 IP LC-MS/MS approach (Figure 4G,H). The ChIP-seq data also show ISWI enrichment in the promoters of the genes that are down-regulated in response to ISWI knockdown (including the active var gene) (Figure 4D). These data, combined with the frequency plot mentioned above (Figure 4C), suggest a direct effect of ISWI binding on transcriptional activation. While ISWI may not be specific to var genes, we believe that the association of ISWI, a putative chromatin remodeler, with accessible intergenic regions of the genome (Figure 4E) and with MORC, ACS, and an ApiAP2 transcription factor (Figure 3C,D) has important implications for regulation of var and other genes and will spawn an interesting new line of research in P. falciparum epigenetics. We have changed any language in the manuscript suggesting that ISWI is specific to var genes.

Thank you for sending us your revised manuscript. We have now heard back from the two reviewers who were asked to evaluate your study. As you will see the reviewers are overall satisfied with the modifications made. They raise, however, a series of -mostly minor- concerns, which should be carefully addressed.

Before we can formally accept your manuscript, we would ask you to address a few remaining editorial issues listed below.

REFEREE REPORTS

Reviewer #1:

This revised manuscript is substantially improved in a number of ways. In particular, inclusion of the control proteomics data from the non-targeted dCas9 line increases confidence in the specificity of the mass spec data, with Figure 3B and 3C clear examples of how the ability to interpret and assign confidence to hits is improved for reviewer and reader alike. Inclusion of the ISWI CHIPseq experiment have also added substantially to the functional validation. All my major concerns have been dealt with, and the authors should be commended for the rapid and comprehensive manner in which they have responded to review, which I believe have improved the manuscript. Only two minor notes to make:

1) It would be helpful to label all of the proteins highlighted by the ISWI pull-down (Fig 3D) on the dCas9 proteomics figures (Fig 3A, 3B) to make clear how the two orthogonal approaches overlap/complement each other.

2) The note made in the response to the reviewers "To be completely honest we have attempted to tag/knockout/knockdown at least a dozen from our study over the past year" is actually hugely useful information for the field. It's highly likely that these essential chromatin-associated proteins are essential and very sensitive to any manipulation that might impair their function, even slightly. I would therefore strongly encourage the authors to include this statement in some form, specifying the genes attempted, in the Discussion, with due caveats about the inefficiencies of Pf transfection etc. It may just save a PhD student somewhere a year of work who might otherwise decide to repeat exactly this strategy....

Reviewer #2:

This is a clearly improved version of the manuscript, including valuable new data and new analyses that greatly assist in the interpretation of the results. As it stands, the main limitation of this study is still that the datasets generated appear to have a lot of nonspecific background (the most common problem for targeted locus immunoprecipitation approaches), with many likely false hits among the list of proteins associated with var genetic elements (and possibly also among the genes that change expression in the ISWI KD). While the authors appear to be reluctant to use more stringent controls, in my opinion it would be more valuable for the field to provide shorter lists of high confidence hits, rather than low confidence long lists (e.g., it is unlikely that >1,000 proteins bind var loci). In spite of this limitation, this is clearly an important study and given the novelty of the approach and the importance of the method described, I do not request a new re-analysis of the data and I only have a few minor suggestions.

-Page 6. TSS is still used to refer to the translation start site (ATG start codon). This may create confusion because TSS is commonly used to refer to Transcription Start Sites (as the authors themselves do in the response letter).

-The highest expression of targeted var genes in the var promoter-targeted dCas9 line is striking (Fig EV2D, top) and should be discussed. Of the 50+ var genes, the 10 more highly expressed appear to be all among the 17 var that are targeted. This is not a random distribution. The authors claim that they don't want to speculate about the reasons for this, but this cannot be ignored. The data available suggest that high expression may be a consequence of dCas9 binding. Possible scenarios that may explain this observation should at least be discussed.

-Page 12. The sentence "As ISWI was one of the most highly enriched factors at the var gene promoter..." may be misleading. The fold enrichment of ISWI at var promoters (either vs intron or vs control) is modest, e.g. it's not in the top 50 of any of the lists of enriched factors.

-Page 13, "strikingly, several proteins identified were also enriched in the var gene promoter-targeted dCas9 IP LC-MS/MS analysis". In fig. 3c, is the overlap between datasets (4 genes present in all the comparisons) higher than expected randomly? The level of overlap may not be considered striking.

Reviewer #1:

This revised manuscript is substantially improved in a number of ways. In particular, inclusion of the control proteomics data from the non-targeted dCas9 line increases confidence in the specificity of the mass spec data, with Figure 3B and 3C clear examples of how the ability to interpret and assign confidence to hits is improved for reviewer and reader alike. Inclusion of the ISWI CHIPseq experiment have also added substantially to the functional validation. All my major concerns have been dealt with, and the authors should be commended for the rapid and comprehensive manner in which they have responded to review, which I believe have improved the manuscript. Only two minor notes to make:

1) It would be helpful to label all of the proteins highlighted by the ISWI pull-down (Fig 3D) on the dCas9 proteomics figures (Fig 3A, 3B) to make clear how the two orthogonal approaches overlap/complement each other.

The proteins shown in Figure 3D have now been highlighted in Figure 3B since this plot compares the var promoter-targeted dCas9 immunoprecipitation to the intron- and non-targeted dCas9 immunoprecipitations.

2) The note made in the response to the reviewers "To be completely honest we have attempted to tag/knockout/knockdown at least a dozen from our study over the past year" is actually hugely useful information for the field. It's highly likely that these essential chromatin-associated proteins are essential and very sensitive to any manipulation that might impair their function, even slightly. I would therefore strongly encourage the authors to include this statement in some form, specifying the genes attempted, in the Discussion, with due caveats about the inefficiencies of Pf transfection etc. It may just save a PhD student somewhere a year of work who might otherwise decide to repeat exactly this strategy....

We have added this information to the Results section (first paragraph on page 13). However, we hope that this will not discourage those who wish to study these genes, as it is possible that we happened to choose a bad PAM site or had bad luck in general with transfections this past year. An alternative method to CRISPR/Cas9 might yield better results, and we are currently attempting the SLI method.

Reviewer #2:

This is a clearly improved version of the manuscript, including valuable new data and new analyses that

greatly assist in the interpretation of the results. As it stands, the main limitation of this study is still that the datasets generated appear to have a lot of nonspecific background (the most common problem for targeted locus immunoprecipitation approaches), with many likely false hits among the list of proteins associated with var genetic elements (and possibly also among the genes that change expression in the ISWI KD). While the authors appear to be reluctant to use more stringent controls, in my opinion it would be more valuable for the field to provide shorter lists of high confidence hits, rather than low confidence long lists (e.g., it is unlikely that >1,000 proteins bind var loci). In spite of this limitation, this is clearly an important study and given the novelty of the approach and the importance of the method described, I do not request a new re-analysis of the data and I only have a few minor suggestions.

-Page 6. TSS is still used to refer to the translation start site (ATG start codon). This may create confusion because TSS is commonly used to refer to Transcription Start Sites (as the authors themselves do in the response letter).

The single instance that TSS is defined and used in the manuscript (first paragraph on page 6) has been removed.

-The highest expression of targeted var genes in the var promoter-targeted dCas9 line is striking (Fig EV2D, top) and should be discussed. Of the 50+ var genes, the 10 more highly expressed appear to be all among the 17 var that are targeted. This is not a random distribution. The authors claim that they don't want to speculate about the reasons for this, but this cannot be ignored. The data available suggest that high expression may be a consequence of dCas9 binding. Possible scenarios that may explain this observation should at least be discussed.

The authors respectfully disagree that higher expression of targeted *var* genes is a consequence of dCas9 binding. On the contrary, usually dCas9 binding to promoter regions interferes with transcription, which is why we wanted to show that this was not the case with our system (Baumgarten et al., 2019; Barcons-Simon et al., 2020). We do believe that higher expression of dCas9-targeted *var* genes might be a consequence of the fact that active promoters are simply more accessible than silent promoters, as has been shown by multiple studies using ATAC-seq (Ruiz et al., 2018; Toenhake et al., 2018). If dCas9 can more easily access its target sequence in euchromatic promoters, perhaps parasites expressing dCas9 that is able to bind to the genome were more readily selected for during transfection. Although we are still uncomfortable speculating about the reason behind this phenomenon, we have added text stating this hypothesis to the Results section (first paragraph on page 8).

-Page 12. The sentence "As ISWI was one of the most highly enriched factors at the var gene promoter..." may be misleading. The fold enrichment of ISWI at var promoters (either vs intron or vs control) is modest, e.g. it's not in the top 50 of any of the lists of enriched factors.

The authors apologize for the confusion, but we are referring to *p*-value, not enrichment ratio. ISWI has the seventh lowest *p*-value in the *var* promoter versus intron comparison. While this sentence has already been changed in response to Reviewer 1's request (unrelated to this request), we have added the word "significantly" to the first sentence of the first paragraph of page 13.

-Page 13, "strikingly, several proteins identified were also enriched in the var gene promoter-targeted dCas9 IP LC-MS/MS analysis". In fig. 3c, is the overlap between datasets (4 genes present in all the comparisons) higher than expected randomly? The level of overlap may not be considered striking.

Using the GeneOverlap R package for testing and visualizing gene overlaps (<https://github.com/shenlab-sinai/geneoverlap>), we found that the overlaps are statistically significant when you consider that there are ~4,700 protein-coding genes in the *P. falciparum* genome

(Otto et al., 2014). The Fisher's exact test gives an odds ratio, representing strength of association. An odds ratio ≤ 1 means no association between two lists. An odds ratio > 1 means there is association. We have emphasized this significance in the Results section (first paragraph of page 13) and added a table showing the odds ratios and p -values as Figure EV3F.

Thank you again for sending us your revised manuscript. We are now satisfied with the modifications made and I am pleased to inform you that your paper has been accepted for publication.

Corresponding Author Name: Jessica M. Bryant
 Journal Submitted to: Molecular Systems Biology
 Manuscript Number: MSB-20-9569RRR